



# Thermodynamic Concepts used in Physical Oceanography

5      Trevor J. McDougall[1, 2]

[1]School of Mathematics and Statistics, University of New South Wales, Sydney, NSW 2052, Australia
[2]This article is based on the 2025 Alfred Wegener Medal lecture "Looking under the hood of Physical Oceanography:
10      Curiosities and Surprises" https://meetingorganizer.copernicus.org/EGU25/sessionprogramme/5775 given at the
European Geosciences Union General Assembly in Vienna, 30[th] April 2025.

*Correspondence to*: Trevor McDougall (Trevor.McDougall@unsw.edu.au)

**Index**





**Abstract.** The thermodynamic concepts that are used in physical oceanography are reviewed, including the several different types of salinity, and how the First Law of Thermodynamics is derived. Different types of temperature are discussed, leading to potential enthalpy and Conservative Temperature, because of the need to accurately quantify the ocean's role in transporting heat. A key aspect of a thermodynamic variable is the extent of its non-conservation when mixing occurs at a given pressure. Methods are presented that quantify the amount of non-conservation of several thermodynamic variables, and these are illustrated in the global ocean. There has been confusion in the literature about the meaning of the salinity and temperature variables carried by ocean models, and here we explain why even in older ocean models that carry the EOS-80 equation of state (rather than TEOS-10), the model's salinity is Preformed Salinity and the model's temperature variable is Conservative Temperature. The thermodynamic reasoning that leads to the concept of neutral surfaces is reviewed, along with thermobaricity, cabbeling, the dianeutral motion caused by the ill-defined nature of neutral surfaces, and Neutral Surface Potential Vorticity.

# 1 Introduction

This review article discusses the thermodynamic concepts that lie behind TEOS-10 (the international thermodynamic equation of seawater - 2010). These thermodynamic concepts have influenced our understanding of the nature of lateral mixing in the ocean and have also led to the change of temperature and salinity variables from being potential temperature and Practical



Salinity under EOS-80 to now being Conservative Temperature and Absolute Salinity. Here we restrict the discussion to thermodynamic concepts applicable to the ocean, while the article by Feistel (2024) is an accessible introduction to many thermodynamic concepts that involve evaporation, precipitation, the transport of the enthalpy of humid air, and the climatic implications of these thermodynamic quantities.

Rainer Feistel realized that all the accurately known measurements of thermodynamic properties of seawater could be incorporated into a Gibbs function from which all the thermodynamic quantities can be derived by mathematical operations such as differentiation. In this way the accurate observational information of one property can inform the evaluation of other properties. The Feistel (2008) paper defines the Gibbs function of seawater, and this Gibbs function has been adopted by TEOS-10.

TEOS-10 defines the thermodynamic properties of not only seawater but also of ice and of humid air. In the present article we do not dwell on the history of how TEOS-10 was derived (this is well covered in Pawlowicz e al., 2012) but rather on the thermodynamic knowledge that explains and justifies the choices made in developing TEOS-10. The Intergovernmental Oceanographic Commission (IOC) recommended the adoption of TEOS-10 in place of the International Equation Of State – 1980 (EOS-80):- see Valladares et al. (2011) and Spall et al. (2013) and resolution XXV-7 at IOC's 25th Assembly in June

2009. Many of the research papers of SCOR/IAPSO Working Group 127 which underpin the TEOS-10 standard are published in the special issue "Thermophysical Properties of Seawater" of *Ocean Science*, Pawlowicz et al. (2012).

The new salinity variables of TEOS-10, Reference Salinity, Absolute Salinity and Preformed Salinity are explained in section 2 below. Since ocean models of both the TEOS-10 and EOS-80 varieties treat their salinity variable as being conservative, it is clear that the salinity variable in ocean models is neither Absolute Salinity nor Reference Salinity (nor is it

Practical Salinity in the case of EOS-80 models). In section 9 we revise the arguments that show that the salinity variable in these models is Preformed Salinity $S_*$ (or $S_*/u_{PS}$ in the case of EOS-80 models). It is now fifteen years since the introduction of TEOS-10, but no attempt has yet been made in ocean models to enable the evaluation of specific volume using Absolute Salinity, and we estimate that meridional overturning transports are currently in error by an estimated 13.5% because of this neglect.

Thermodynamic theory begins with the Fundamental Thermodynamic Relationship, and following from this, the evolution equations for total energy and the First Law of Thermodynamics can be derived (section 3). Oceanographic practice traditionally assumes that potential temperature is both a "potential" variable and a "conservative" variable. Under TEOS-10 this use of a temperature variable that is both a "potential" and a "conservative" variable can continue, but with the new variable, Conservative Temperature. With this change TEOS-10 has brought an improvement by a factor of a hundred

compared with the non-conservative production of potential temperature.

Section 8 discusses the amount by which various thermodynamic variables are non-conservative in the ocean. One measure of such non-conservation is the vertical integral over the full ocean depth of the non-conservation due to the estimated mixing in the ocean, expressed as a surface heat flux. This shows that the non-conservation of Conservative Temperature is usually





less than 1 mW m$^{-2}$ while that of potential temperature is a hundred times larger, being approximately the same magnitude
as the geothermal heat flux.

The reasoning in terms of buoyant restoring forces that leads to the notion of the neutral tangent plane is discussed in
section 10. This leads into the dianeutral advection processes thermobaricity and cabbeling, and also to the path-dependent
nature of neutral surfaces. The final chapter (chapter 13) discusses the influence of the nonlinear nature of the equation of
state on the meaning of planetary potential vorticity.

## 2 Absolute Salinity, Preformed Salinity, Reference Salinity and Practical Salinity

The derived thermodynamic property of seawater of most importance in physical oceanography is the specific volume, or
density. Between 1980 and 2010 specific volume was evaluated using the EOS-80 equation which was a function of Practical
Salinity, in-situ temperature and pressure (Fofonoff, 1985). Measurements of temperature and pressure at sea can be made
relatively accurately compared to those of Practical Salinity, and since 1948 the International Association for the Physical
Sciences of the Oceans (IAPSO) has encouraged the use of ampules of Standard Seawater of known conductivity ratio and
known Practical Salinity to assist in measuring Practical Salinity at sea as accurately as possible (Smythe-Wright et al., 2019).
This service is now performed by a private company, Ocean Scientific International Limited (Jenkins and Williams, 2025)
which supplies bottles of IAPSO Standard Seawater. With great care it is possible to measure Practical Salinity with an
accuracy of 0.003 (two standard deviations), and continuing research is encouraged to ensure this accuracy is maintained into
the future, and if possible, improved (Uchida et al., 2025). Two possible routes for improvements are via the use of vibrating
beam densimeters that measure density in an SI-traceable manner (Wright et al., 2011), and refractive index sensors (Uchida
et al. (2019), Li et al. (2023), Yang et al. (2024), Bai et al. (2025), Zhao et al. (2025)), but these possible future avenues for
improvement will not be further mentioned in this article.

Practical Salinity is based on a measurement of the electrical conductivity ratio of seawater, corrected for the temperature
and pressure of the seawater sample. TEOS-10 (the International Thermodynamic Equation of Seawater – 2010) recognises
that the relative concentrations of the constituents of sea salt in seawater vary throughout the ocean, and these variations
influence the electrical conductivity differently to how these same variations affect specific volume. Pawlowicz (2010), Wright
*et al*. (2011) and IOC *et al*. (2010) discuss the several contenders for the title of the "absolute salinity" of seawater, namely
"Solution Salinity", "Added-Mass Salinity", and "Density Salinity". The paper of Wright *et al*. (2011) presents a clear and
readable account of this difficult subject. Under TEOS-10 the capitalized words Absolute Salinity and the symbol $S_A$ are
reserved for "Density Salinity" such as can be deduced using laboratory measurements with a vibrating beam densimeter. That
is, Absolute Salinity is defined to be that salinity that when used as the salinity argument of the TEOS-10 expression for
specific volume, gives the actual specific volume of the seawater sample. Absolute Salinity is expressed on the Reference-
Composition Salinity Scale of Millero et al. (2008) and is very close to being the mass fraction of sea salt (non-water) in a
seawater sample.





TEOS-10 concentrates on four different salinity variables, namely Practical Salinity $S_P$, Reference Salinity $S_R$, Absolute Salinity $S_A$, and Preformed Salinity $S_*$. Underlying these salinity variables is the paper of Millero et al. (2008) which defined the composition of Reference-Composition Seawater as a table of exact mole fractions of the main chemical constituents of seawater. This table defines the best estimate of the composition of Standard Seawater, which is seawater from the surface

waters of a certain region of the North Atlantic. For seawater of Reference Composition, its Absolute Salinity is related to Practical Salinity by

$$S_R = u_{PS} S_P \quad \text{where} \quad u_{PS} \equiv (35.16504/35) \text{ g kg}^{-1} \tag{1}$$

where $S_R$ is called the Reference Salinity. If the composition of a seawater sample is different to that of Reference-Composition Seawater, then its Reference Salinity can still be calculated using Eq. (1) (this is the Absolute Salinity of the

sample under the assumption that the sample is of Reference Composition) and the sample's Absolute Salinity is calculated as

$$S_A = S_R + \delta S_A, \tag{2}$$

where $\delta S_A$ is called the Absolute Salinity Anomaly and is usually evaluated from the computer software of the Gibbs SeaWater Oceanographic Toolbox (McDougall and Barker, 2011). This approach for estimating the Absolute Salinity Anomaly is actually based on laboratory measurements of the density of seawater samples collected from around the world's oceans as

described in McDougall *et al.* (2012). However, if open ocean measurements are available of the Total Alkalinity, Dissolved Inorganic Carbon, and the nitrate and silicate concentrations, then an alternative formula is available to calculate Absolute Salinity according to the expression,

$$(S_A - S_R)/(\text{g kg}^{-1}) = (55.6 \, \Delta\text{TA} + 4.7 \, \Delta\text{DIC} + 38.9 \, \text{NO}_3^- + 50.7 \, \text{Si(OH)}_4)/(\text{mol kg}^{-1}) \,. \tag{3}$$

This approach was developed by Pawlowicz *et al.* (2011) using a chemical model of the electrical conductivity and density of

seawater. This equation is written in terms of the values of the nitrate and silicate concentrations in the seawater sample (measured in mol kg$^{-1}$), the differences $\Delta$TA and $\Delta$DIC, between the Total Alkalinity (TA) and Dissolved Inorganic Carbon (DIC) of the sample and the corresponding values of our best estimates of TA and DIC in Standard Seawater. For Standard Seawater our best estimates of TA and DIC are 0.0023 ($S_P/35$) mol kg$^{-1}$ and 0.00208 ($S_P/35$) mol kg$^{-1}$ respectively (see the discussion in Wright *et al.* (2011)).

Preformed Salinity $S_*$ is designed to be a conservative salinity variable which is unaffected by biogeochemical activity in the ocean; it is defined as Absolute Salinity minus the contributions of biogeochemical processes to Absolute Salinity. Based on the work of Pawlowicz et al. (2011) the difference between Absolute Salinity and Preformed Salinity is approximately proportional to the difference between Absolute Salinity and Reference Salinity, with the proportionality constant being 1.35. That is, $(S_A - S_*) \approx 1.35(S_A - S_R)$, and this is illustrated in Figure 1.





This section has just scratched the surface of this subject of salinity. The interested reader will find more details on the various salinity variables, in previous review articles on the thermodynamics of seawater (McDougall et al., 2013 and Feistel, 2018), and in the Wright et al. (2011) paper which is the most comprehensive discussion of these salinity issues.

## 3 The First Law of Thermodynamics

### 3.1 The Fundamental Thermodynamic Relationship

The fundamental thermodynamic relationship (FTR) is the following differential relationship between the total derivatives of internal energy, $u$, specific volumes, $v$, entropy, $\eta$, and Absolute Salinity $S_A$,

$$\mathrm{d}h - v\mathrm{d}P \ = \ \mathrm{d}u + P\mathrm{d}v = \ T\mathrm{d}\eta + \mu\mathrm{d}S_A. \tag{4}$$

The first part of this equation serves to introduce the specific enthalpy, $h \equiv u + Pv$, defined as the sum of internal energy and the product of absolute pressure, $P$, and specific volume, $v$. The total differentials in the FTR represent reversible differences between equilibrium states that are separated by vanishingly small differences in state variables.

We tend to regard Absolute Salinity, $S_A$, in-situ absolute temperature, $T$, and absolute pressure, $P$, as physical properties that can be measured in the ocean, and in terms of these variables, enthalpy, $h$, internal energy, $u$, specific volume, $v$, entropy, $\eta$, and relative chemical potential, $\mu$, are state variables that are functions of $(S_A, T, P)$ and do not depend on how a system evolves through time or space from one equilibrium thermodynamic state to another.

In order to understand the FTR consider a fluid parcel that is not exchanging any heat or salt with its surroundings, and nor is there is any internal dissipation of kinetic energy. In this situation we know that neither the salinity nor the entropy of the fluid parcel change and we say that the flow is isohaline ($\mathrm{d}S_A = 0$) and isentropic ($\mathrm{d}\eta = 0$). In this situation any change in volume results in a change in the internal energy according to $\mathrm{d}u = -P\mathrm{d}v$, while any change in pressure causes the enthalpy to change according to $\mathrm{d}h = v\mathrm{d}P$. Consider now supplying a small amount of heat to the system (one way of doing this is to

dissipate some kinetic energy) while not having any exchange of mass (so the salinity is unchanged). The amount of heat supplied is equal to $T\mathrm{d}\eta$ (this follows from the Second Law of Thermodynamics) and if the process occurs at constant pressure this is equal to the change in enthalpy, $\mathrm{d}h$, while if the heating occurs at constant volume, $T\mathrm{d}\eta$ will equal $\mathrm{d}u$. The $\mu\mathrm{d}S_A$ term in (1), being the product of $\mathrm{d}S_A$ and the relative chemical potential of sea salt in seawater, $\mu$, represents for example, the influence of the salinity change on enthalpy (if the change in salinity occurs at constant pressure and entropy). A more

extensive discussion of the Fundamental Thermodynamic Relationship and the First Law of Thermodynamics in an oceanographic context can be found in section 1 of McDougall et al. (2023).

### 3.2 The Evolution Equation of Total Energy

The First Law of Thermodynamics expresses how fast the enthalpy, internal energy and entropy of a fluid parcel change when the fluid parcel is heated. The route by which the First Law of Thermodynamics is developed for a fluid is not obvious and is





not routinely or consistently treated in oceanographic textbooks. The First Law of Thermodynamics cannot be derived directly but rather follows from the evolution equation of total energy, $E = u + 0.5\mathbf{u} \cdot \mathbf{u} + \Phi$, being the sum of internal energy, $u$, kinetic energy, $0.5\mathbf{u} \cdot \mathbf{u}$, and gravitational potential energy, $\Phi$. The derivation of the evolution equation of total energy in this section follows closely Appendix B of the TEOS-10 Manual (IOC et al., 2010) which in turn follows the clearest text on the subject, Landau and Lifshitz (1959).

We first construct the evolution equation for mechanical energy, which is the sum of kinetic energy, $0.5\mathbf{u} \cdot \mathbf{u}$, and gravitational potential energy, being the integral of the gravitational acceleration with respect to height. This evolution equation is derived in detail in quality fluid dynamics textbooks (e. g. Batchelor, 1970)

$$(\rho[0.5\mathbf{u} \cdot \mathbf{u} + \Phi])_t + \nabla \cdot (\rho\mathbf{u}[0.5\mathbf{u} \cdot \mathbf{u} + \Phi]) = \rho\, \mathrm{d}[0.5\mathbf{u} \cdot \mathbf{u} + \Phi]/\mathrm{d}t = -\mathbf{u} \cdot \nabla P + \nabla \cdot (\rho v^{\mathrm{visc}}\nabla[0.5\mathbf{u} \cdot \mathbf{u}]) - \rho\varepsilon. \quad (5)$$

Here $v^{\mathrm{visc}}$ is the dynamic viscosity, $\varepsilon$ is the dissipation of kinetic energy, and the density, $\rho$, is the reciprocal of specific 200   volume, $v$.

  The next equation to be derived is the evolution equation for total energy, $E \equiv u + 0.5\mathbf{u} \cdot \mathbf{u} + \Phi$, being the sum of internal energy, kinetic energy and gravitational potential energy. This evolution equation is derived in two steps (IOC et al., 2010, following Landau and Lifshitz, 1959). The first step is to consider a situation in which there are no molecular fluxes of heat, salt or momentum and no dissipation of kinetic energy. In this situation a fluid parcel's entropy and salinity do not change 205   with time so that we see from the FTR that the material derivative of internal energy, $\mathrm{d}u/\mathrm{d}t$, is equal to $-P\mathrm{d}v/\mathrm{d}t$. When $\rho\mathrm{d}u/\mathrm{d}t$ and $-\rho P\mathrm{d}v/\mathrm{d}t$ (which, via the continuity equation can be written as $-P\nabla \cdot \mathbf{u}$) are added to the left- and right-hand sides respectively of Eq. (5), on finds that $\rho\, \mathrm{d}E/\mathrm{d}t = -\nabla \cdot (P\mathbf{u})$, which is the adiabatic and non-viscous version of the evolution equation for total energy. The second step is to add two forcing terms to the right-hand side of this equation, one representing the influence of molecular and radiative heat fluxes and the other representing the effect of viscosity on the 210   evolution of total energy. In order to ensure that these two terms contribute no net sources or sinks to the total energy in the interior of the fluid, the forcing terms are imposed in the form of the divergence of fluxes, obtaining

$$(\rho E)_t + \nabla \cdot (\rho\mathbf{u}E) = \rho\, \mathrm{d}E/\mathrm{d}t = -\nabla \cdot (P\mathbf{u}) - \nabla \cdot \mathbf{F}^Q + \nabla \cdot (\rho v^{\mathrm{visc}}\nabla[0.5\mathbf{u} \cdot \mathbf{u}]). \qquad (6)$$

  These two equations are the key to deriving and understanding the First Law of Thermodynamics. The above evolution equations for mechanical energy and total energy are the unaveraged equations for the instantaneous flow being forced by the 215   molecular diffusion of heat, salt and momentum. These equations do not represent the mean flow after averaging over turbulent motions or over the finite volumes of grid boxes in numerical models. Such equations are the result of additional averaging processes, some of which we discuss below.

### 3.3 The First Law of Thermodynamics

The First Law of Thermodynamics is obtained by subtracting the evolution equation of mechanical energy, Eq. (5) from the 220   evolution equation of total energy, Eq. (6) obtaining (after again using the $\rho\, \mathrm{d}v/\mathrm{d}t = \nabla \cdot \mathbf{u}$ form of the continuity equation)





$$\rho(\mathrm{d}h/\mathrm{d}t - v\,\mathrm{d}P/\mathrm{d}t) = \rho(\mathrm{d}u/\mathrm{d}t + P\,\mathrm{d}v/\mathrm{d}t) = \rho(T\mathrm{d}\eta/\mathrm{d}t + \mu\,\mathrm{d}S_A/\mathrm{d}t) = -\nabla \cdot \mathbf{F}^Q + \rho\varepsilon. \tag{7}$$

The FTR, Eq. (4), proves the equivalence of the first three parts of this equation. This First Law of Thermodynamics can be written as the following evolution equations for internal energy and for enthalpy,

$$(\rho u)_t + \nabla \cdot (\rho \mathbf{u}u) = \rho\,\mathrm{d}u/\mathrm{d}t = -P\nabla \cdot \mathbf{u} - \nabla \cdot \mathbf{F}^Q + \rho\varepsilon. \tag{8}$$

$$(\rho h)_t + \nabla \cdot (\rho \mathbf{u}h) = \rho\,\mathrm{d}h/\mathrm{d}t = \mathrm{d}P/\mathrm{d}t - \nabla \cdot \mathbf{F}^Q + \rho\varepsilon. \tag{9}$$

In the above development we have ignored a small term due to the non-conservation of Absolute Salinity mostly caused by the remineralization of organic matter in the ocean. While this term is important in the salinity evolution equation, it can be shown to be negligible in the First Law of Thermodynamics (see Appendix A.21 of IOC et al., 2010).

The First Law of Thermodynamics (7) contains the divergence of the molecular flux of heat, $\nabla \cdot \mathbf{F}^Q$, as a forcing term, and

the evolution equation for salinity has the corresponding term $\nabla \cdot \mathbf{F}^S$ representing the divergence of the molecular diffusion of salt. Thermodynamic theory dictates that these molecular fluxes have the following forms

$$\mathbf{F}^S = A\nabla(-\mu/T) + B\nabla(1/T), \tag{10}$$

$$\mathbf{F}^Q = B\nabla(-\mu/T) + C\nabla(1/T). \tag{11}$$

As discussed in Appendix B of the TEOS-10 Manual (IOC et al., 2010), the Second Law of Thermodynamics constraint that

the production of entropy is always non-negative requires that both $A$ and $C$ are positive and that $B^2 < AC$. When the gradient of $-\mu/T$ in Eq. (10) is expanded in terms of gradients of Absolute Salinity, temperature and pressure, one finds that an ocean in which there were no molecular diffusion of either heat or salt would have spatially constant values of both in-situ temperature (and hence non-uniform entropy) and chemical potential $\mu$, requiring a vertical gradient of Absolute Salinity of approximately $3\ \mathrm{g\ kg}^{-1}$ per 1000m in the vertical.

Here we immediately note an important feature of the First Law in the form Eq. (9), namely that when fluid parcels are mixed at constant pressure, $\mathrm{d}P/\mathrm{d}t$ is zero, and the enthalpy of the final parcel is the sum of the initial two enthalpies except for the heating effect of the dissipation of kinetic energy, $\varepsilon$. This property of enthalpy applies for turbulent mixing between fluid parcels and is possibly the most important feature of thermodynamics that oceanographers need to know, as it is the first of three physical arguments that together, justify the usefulness of potential enthalpy and Conservative Temperature. This is

discussed in more detail below.

## 4 "Potential" and "Conservative" variables

### 4.1 "Potential" variables

A variable is called a "potential" variable if its value in a fluid parcel is unchanged when the parcel's pressure is changed without any exchange of heat or matter. That is, a "potential" variable is unchanged when pressure is varied in an adiabatic



and isohaline manner. Examples of potential variables are entropy, potential temperature, potential density, and potential enthalpy.

We now consider an adiabatic and isohaline pressure change of 1000 dbar ($10^7$ Pa) to illustrate the sensitivity of several other variables (that are not "potential" variables) to pressure changes. For such a pressure change the in-situ temperature, $T$, changes by ~0.1°C (usually an increase in temperature, but for very cold temperatures where the thermal expansion coefficient

is negative, the temperature change is negative), while internal energy, $u$, increases by ~$10^2$ J kg$^{-1}$ which is the same change in internal energy as caused by ~0.025°C of warming. Enthalpy has a much larger sensitivity to pressure, increasing by ~$10^4$ J kg$^{-1}$ for the same adiabatic and isohaline increase in pressure of 1000dbar; this increase in enthalpy being equivalent to that caused by ~2.4°C of warming (see Figure 3 of McDougall et al., 2021a). The total energy $E$ is as sensitive to such an adiabatic and isohaline change in pressure as is enthalpy, but with the opposite sign, so that an adiabatic and isohaline increase

in hydrostatic pressure of 1000dbar results in a decrease of $E$ of ~$10^4$ J kg$^{-1}$, being the same change as that caused by ~2.4°C of cooling.

## 4.2 "Conservative" variables

A "conservative" variable has the property that when two fluid parcels are brought together and mixed while not exchanging heat or matter with the environment, the total amount of the variable in the final state is the sum of the amounts contained in

the original two fluid parcels. A conservative variable, $C$, obeys the evolution equation

$$(\rho C)_t + \nabla \cdot (\rho \mathbf{u} C) = \rho \, dC/dt = -\nabla \cdot \mathbf{F}^C, \tag{12}$$

where $\mathbf{F}^C$ is the flux of property $C$ caused by molecular diffusion. This restriction to the flux being a molecular flux is important and is discussed in detail in sections A8 and A9 of IOC et al. (2010). It is not sufficient that the right-hand side of Eq. (12) is simply the divergence of a flux as will become obvious when we discuss the non-conservative nature of total energy.

Absolute Salinity, $S_A$, is not a conservative variable because of the remineralization of organic matter, which is denoted by the source term, $\mathcal{S}$, in the evolution equation of Absolute Salinity,

$$(\rho S_A)_t + \nabla \cdot (\rho \mathbf{u} S_A) = \rho \, dS_A/dt = -\nabla \cdot \mathbf{F}^S + \mathcal{S}. \tag{13}$$

Preformed Salinity, $S_*$, is defined to be the absolute salinity that would occur in the ocean if there were no remineralization of organic matter (Wright et al., 2011). Preformed Salinity is a conservative variable, obeying the evolution equation

$$(\rho S_*)_t + \nabla \cdot (\rho \mathbf{u} S_*) = \rho \, dS_*/dt = -\nabla \cdot \mathbf{F}^S, \tag{14}$$

where $\mathbf{F}^S$, is the molecular diffusive flux of salt.

Here we expand on the meaning of the "conservative" property, and why it is important that the flux whose divergence appears on the right-hand side of Eq. (12) is the molecular diffusion flux of the property. We will do so by addressing the extent of the conservation of two of the variables we have discussed above, namely potential enthalpy, $h^m$, and total energy,





$E$. Consider two fluid parcels that have been moved to be next to each other at the same pressure $P^m$. The two parcels turbulently mix together at this pressure, and we consider the variable called potential enthalpy referenced to the pressure $P^m$. In terms of the enthalpy function $\hat{h}(S_A, \Theta, P)$, potential enthalpy with respect to the reference pressure $P^m$ is $h^m = \hat{h}(S_A, \Theta, P^m)$. Because $h^m$ is a potential quantity it has the advantage that the potential enthalpy of each parcel does not change during the adiabatic and isohaline movements that bring the parcels to be adjacent to each other. Now we allow the two parcels to turbulently mix at the constant pressure $P^m$. During this mixing process at the pressure $P^m$ the evolution equation of enthalpy, Eq. (9), applies and it can be written as

$$(\rho h^m)_t + \nabla \cdot (\rho \mathbf{u} h^m) = \rho \, dh^m/dt = -\nabla \cdot \mathbf{F}^Q + \rho\varepsilon. \qquad \text{at } P = P^m \qquad (15)$$

When Eq. (15) is spatially and temporally integrated over a moving and contracting volume in which a mixing event is occurring, the Leibnitz differentiation of the volume integral of $\rho h^m$ ensures that the relevant surface velocity that affects the volume-integrated properties is the velocity *through* this moving boundary, the dia-surface velocity, $\mathbf{u}^{\text{dia}}$. This is proven by considering the time differentiation of the volume integral of the total amount of $h^m$-substance in the volume, as on the left-hand side of Eq. (16) below. The last term on the right-hand side of the first line of this equation arises from the fact that the boundary is moving through space, with $\mathbf{u}^{\text{boundary}}$ being the velocity of the bounding surface of the volume. In the second line of Eq. (16), Eq. (15) has been used to replace the temporal derivative term, $(\rho h^m)_t$, that appears in the first line, while in the third line we convert two of the three volume integrals into boundary area integrals using the divergence theorem, and then use the definition of the dia-surface velocity, $\mathbf{u}^{\text{dia}} = \mathbf{u} - \mathbf{u}^{\text{boundary}}$.

$$\frac{\partial}{\partial t}\left(\int_V \rho h^m dV\right) = \int_V (\rho h^m)_t \, dV + \int_S \rho h^m \mathbf{u}^{\text{boundary}} \cdot d\mathbf{S}$$

$$= -\int_V \nabla \cdot (\rho h^m \mathbf{u} + \mathbf{F}^Q) \, dV + \int_S \rho h^m \mathbf{u}^{\text{boundary}} \cdot d\mathbf{S} + \int_V \rho\varepsilon \, dV$$

$$= -\int_S \left(\rho h^m \mathbf{u}^{\text{dia}} + \mathbf{F}^Q\right) \cdot d\mathbf{S} + \int_V \rho\varepsilon \, dV. \qquad (16)$$

With the control volume extending into quiescent fluid which is not involved in the turbulent mixing, and in the absence of molecular mixing, $\left(\rho h^m \mathbf{u}^{\text{dia}} + \mathbf{F}^Q\right)$ is zero on the boundary of the control volume so that the volume integral of potential enthalpy is changed only by the volume integral of the dissipation of turbulent kinetic energy per unit volume, $\rho\varepsilon$. We conclude that apart from the so-called "Joule heating" of the dissipation of kinetic energy, potential enthalpy with reference pressure $P^m$ is conserved when turbulent mixing of fluid parcels takes place at this pressure.

This almost conservative behaviour of potential enthalpy $h^m$ is now contrasted with the non-conservative behaviour of total energy $E$. Performing the same type of Leibnitz differentiation of the volume integral of the amount of total energy we find the first line of Eq. (17) just as we did in Eq. (16), and the second line of (17) follows after using (6).





$$\frac{\partial}{\partial t}\left(\int_V \rho E \, dV\right) = \int_V (\rho E)_t \, dV + \int_S \rho E \mathbf{u}^{\text{boundary}} \cdot d\mathbf{S}$$


$$= -\int_V \nabla \cdot (\rho E \mathbf{u} + P\mathbf{u} + \mathbf{F}^Q - \rho v^{\text{visc}}\nabla[0.5\mathbf{u}\cdot\mathbf{u}]) \, dV \; + \int_S \rho E \mathbf{u}^{\text{boundary}} \cdot d\mathbf{S}$$

$$= -\int_S \left(\rho E \mathbf{u}^{\text{dia}} + \mathbf{F}^Q - \rho v^{\text{visc}}\nabla[0.5\mathbf{u}\cdot\mathbf{u}]\right)\cdot d\mathbf{S} - \int_S P\mathbf{u}\cdot d\mathbf{S}. \qquad (17)$$

In the absence of molecular mixing $\left(\rho E \mathbf{u}^{\text{dia}} + \mathbf{F}^Q - \rho v^{\text{visc}}\nabla[0.5\mathbf{u}\cdot\mathbf{u}]\right)$ is zero on the boundary of our same control volume, but now there is a remaining surface integral of $-P\mathbf{u}$ over the surface of the control volume. This surface integral appears because the fluxes whose divergence appears on the right-hand side of the evolution equation for $E$, Eq. (6), are not all molecular fluxes; one of them is the divergence of $-P\mathbf{u}$. Other serious drawbacks of total energy $E$ as far as use as a physical
oceanographic variable are that (1) it is not a thermodynamic variable (since it is not a function of only salinity, temperature and pressure), and (2) it is not a "potential" variable; see section 4.1 above where it was shown that total energy is very sensitive to adiabatic and isohaline changes in pressure, with an adiabatic and isohaline increase in pressure of 1000dbar causing the same decrease in total energy as would be caused by ~2.4°C of cooling at the original pressure.

## 5 In-situ temperature and the adiabatic lapse rate

Here we discuss in-situ temperature and the rate at which it changes with pressure, even when the pressure changes occur adiabatically and without change in salinity. In particular, we expose the physical cause of this adiabatic change in temperature (the adiabatic lapse rate).

From the Fundamental Thermodynamic Relationship (FTR), Eq. (4), we find that

$$T = \left.\frac{\partial h}{\partial \eta}\right|_{S_A, P} = \hat{h}_\Theta / \hat{\eta}_\Theta, \qquad \text{that is,} \qquad \frac{T}{\hat{h}_\Theta} = \frac{1}{\hat{\eta}_\Theta}, \qquad (18)$$

While the product of temperature and entropy, $T\eta$, is independent of the scaling of the temperature variable (for example, the use of Kelvin or Fahrenheit scales for absolute temperature), the absolute temperature itself is subject to such a choice of scale. Hence when discussing the adiabatic lapse rate, namely the temperature changes due to adiabatic and isentropic changes in pressure, we consider the variation of $\ln(T)$ instead of $T$. Because entropy is unchanged by adiabatic and isohaline changes in pressure, $\hat{\eta}_\Theta$ in Eq. (18) is a function of $S_A$ and $\Theta$ but not of pressure, so that


$$\left.\frac{\partial \ln(T)}{\partial P}\right|_{S_A, \Theta} = \left.\frac{\partial \ln(\hat{h}_\Theta)}{\partial P}\right|_{S_A, \Theta} = \frac{\hat{h}_{\Theta P}}{\hat{h}_\Theta} = \frac{\hat{v}_\Theta}{\hat{h}_\Theta} = \left.\frac{\partial v}{\partial h}\right|_{S_A, P}. \qquad (19)$$

This expression can also be found from the following physical explanation of the adiabatic lapse rate using enthalpy (as opposed to using internal energy as discussed below). When the pressure on a fluid parcel is increased isentropically and at





constant salinity by $\delta P$ its specific enthalpy increases by $\delta h = v\delta P$. Now considering enthalpy to be in the functional form $h(S_A, T, P)$, the increase in enthalpy is also $\left.\frac{\partial h}{\partial T}\right|_{S_A,P} \delta T + \left.\frac{\partial h}{\partial P}\right|_{S_A,T} \delta P$, and since in this case $h_P = v - Tv_T$, equating these two

expressions for the enthalpy change of the fluid parcel shows that $\left.\frac{\partial \ln(T)}{\partial P}\right|_{S_A,\Theta}$ is equal to $v_T/h_T$, which is the same as Eq. (19).

Since $\hat{h}_\Theta = c_p^0 + \int_{P_0}^{P} \hat{v}_\Theta(P')\,dP'$, the only physical property of the fluid that appears in the expression (R.02) for $\left.\frac{\partial \ln(T)}{\partial P}\right|_{S_A,\Theta}$ is information about thermal expansion, $\hat{v}_\Theta$, and Eq. (19) is completely independent of the compressibility of seawater. Moreover, of the two properties enthalpy and entropy, Eq. (19) depends only on $\hat{h}(S_A, \Theta, P)$ and is completely independent of $\hat{\eta}(S_A, \Theta)$. This independence of $\hat{\eta}(S_A, \Theta)$ is consistent with the ratio of the absolute in-situ and potential

temperatures, $(T_0 + t)/(T_0 + \theta) = \hat{h}_\Theta/c_p^0$, (see McDougall et al., 2021a) also being independent of the expression for entropy. Also, as expected, both this expression for the ratio of absolute in-situ and potential temperatures, and Eq. (19), are independent of the four arbitrary constants that appear in the Gibbs function of seawater (see section 6).

The expression (19) invites one to think of supplying a small amount of heat to a seawater parcel at constant salinity and pressure thereby increasing its enthalpy by $\delta h$, with the resulting changes in specific volume and in entropy being $\delta v$ and $\delta \eta$.

The absolute temperature is $\delta h/\delta \eta$ while the rate at which $\ln(T)$ adiabatically and isentropically increases with pressure is $\delta v/\delta h$.

In almost all atmospheric textbooks, and in oceanographic textbooks published before 2003, whenever a physical explanation of the adiabatic lapse rate is attempted, it is said to be due to the work done on a fluid parcel as its volume changes in response to a change in pressure. This explanation, if true, would have the adiabatic lapse rate being proportional to the

product of pressure and the adiabatic compressibility of the fluid parcel, but this is not the case. McDougall and Feistel (2003) showed that the adiabatic lapse rate of seawater is quite independent of, and is unrelated to, the change in internal energy that a seawater parcel experiences when its pressure is changed. The increase in internal energy, $\delta u$, of a seawater parcel due to an adiabatic and isohaline change in pressure, $\delta P$, is $(Pv\kappa)\delta P$, (where $\kappa$ is the adiabatic compressibility) but the adiabatic lapse rate is not simply $(P\kappa v)$ divided by a straightforward specific heat capacity as one would expect from the traditional

explanation in textbooks. Rather, if this explanation were to be pursued correctly then the relevant "heat capacity" in the denominator would be evaluated at constant salinity and entropy, namely $\left.\frac{\partial u}{\partial T}\right|_{S_A,\eta}$, which for a liquid can tend to infinity and can also be negative.

To isolate what is wrong with this traditional explanation of the adiabatic lapse rate, consider internal energy in the functional form $u(S_A, T, P)$, so that the increase in internal energy of the above parcel, $\delta u = (Pv\kappa)\delta P$, can also be written as

$\left.\frac{\partial u}{\partial T}\right|_{S_A,P} \delta T + \left.\frac{\partial u}{\partial P}\right|_{S_A,T} \delta P$. Equating these two expressions for the increment of internal energy shows that the adiabatic lapse rate $\Gamma$ can be expressed as





$$\frac{\Gamma}{T} = \left. \frac{\partial \ln(T)}{\partial P} \right|_{S_A, \Theta} = \frac{\left( P v \kappa - \left. \frac{\partial u}{\partial P} \right|_{S_A, T} \right)}{T \left. \frac{\partial u}{\partial T} \right|_{S_A, P}} = \frac{\left( P v \kappa - \left. \frac{\partial u}{\partial P} \right|_{S_A, T} \right)}{T \left( c_p - P \left. \frac{\partial v}{\partial T} \right|_{S_A, P} \right)}. \tag{20}$$

This is the correct expression for the adiabatic lapse rate based on the traditional adiabatic and isentropic change of pressure and internal energy of a fluid parcel, although the expression is rather unwieldy. For a liquid $\left( c_p - P \left. \frac{\partial v}{\partial T} \right|_{S_A, P} \right)$ in the

denominator of this expression is different to $c_p$ by no more than 0.1% while for a perfect diatomic gas $\left( c_p - P \left. \frac{\partial v}{\partial T} \right|_{S_A, P} \right) =$

$\frac{5}{7} c_p$ . The error in the traditional textbook explanation of the adiabatic lapse rate is the neglect of the $\left. \frac{\partial u}{\partial P} \right|_{S_A, T}$ term in the numerator of Eq. (20); the term which represents the change in internal energy with pressure at fixed in-situ temperature and salinity. The key difference between a perfect gas and a liquid is that in the case of a perfect gas $u_P|_{S_A, T}$ is zero whereas for a liquid this term is of leading order and is usually much larger than $(P v \kappa)$. The result is that for a perfect gas the traditional

physical explanation in textbooks leads to the correct expression for the adiabatic lapse rate, albeit by incorrect reasoning, while for a liquid the incorrect reasoning would lead to estimates of the adiabatic lapse rate than are often too small by a factor of more than a hundred and can even have the wrong sign (for cool fresh seawater where the thermal expansion coefficient is negative).

## 6 Conservative Temperature

Three different temperatures are in common use in physical oceanography: in-situ temperature, potential temperature and Conservative Temperature. Here we revise previous variables that have been proposed to represent the heat content per unit mass of seawater before outlining the physical intuition that led to considering potential enthalpy and Conservative Temperature for this purpose.

### 6.1 Prior approximations to ocean heat content

Prior to Conservative Temperature being adopted by the oceanographic community in 2010, the usual heat-like variable whose net meridional flux in the ocean was compared to the corresponding air-sea heat flux was potential temperature multiplied by a fixed isobaric heat capacity. Bacon and Fofonoff (1996) had advocated for a different measure of heat content in physical oceanography, namely, $c_p(S_A, \theta, P^0) \theta$, being potential temperature multiplied by the isobaric heat capacity that the seawater parcel would have if moved adiabatically and isentropically to the sea surface pressure, but it turns out that $c_p(S_A, \theta, P^0) \theta$ is

no more accurate as a measure of the heat content of a fluid parcel that is potential temperature itself (McDougall, 2003). Warren (1999) subsequently proposed the use of internal energy, $u$, but, as explained in section 7 below, internal energy is not conserved when fluid parcels mix, with the non-conservation being due to the work done on the fluid parcel by the environment's pressure as the parcel's volume reduces due to cabbeling. Warren (1999) also proposed $\langle c_p \rangle \theta$ as an



approximation to internal energy, where $\langle c_p \rangle$ is the average value of the isobaric heat capacity evaluated at the sample's

salinity, the sea surface pressure, and over a range of potential temperatures between zero Celsius and the parcel's potential

temperature $\theta$. McDougall (2003) showed that $\langle c_p \rangle \theta$ is not a particularly good approximation to internal energy, but rather

is approximately equal to $h(S_A, \theta, P^0) - h(S_A, 0°C, P^0)$, that is, to the potential enthalpy of the fluid parcel minus the potential

enthalpy of a fluid parcel at the same salinity but at zero Celsius temperature. As it turns out, this second option raised by

Warren (1999) would have been a good option if it had not been for the second term, $h(S_A, 0°C, P^0)$.

This short discussion illustrates that several authors have searched for a heat-like variable whose transport in the ocean can

be accurately compared with the air-sea heat flux. We now outline the motivation that lies behind why potential enthalpy (and

hence Conservative Temperature, since it is defined to be proportional to potential enthalpy) was thought to be worth

considering as an approximation to the heat content per unit mass of seawater.

### 6.2 The motivation underlying Conservative Temperature

An ideal oceanographic heat-like variable would have the following three attributes. First, the air-sea flux of heat would be

proportional to the air-sea flux of the variable. Second, the variable would be unchanged by adiabatic and isohaline changes

in pressure; that is, the variable would be a "potential" variable. Third, the variable would be conserved when turbulent mixing

occurred in the ocean interior; that is, the ideal heat-like variable would be a "conversative" variable. The motion and mixing

of fluid parcels in the ocean can be regarded as a sequence of adiabatic and isohaline displacements followed by turbulent

mixing events, so that if a heat-like variable could be found that possessed these three attributes, then its depth-integrated

horizontal fluxes could be accurately compared with the air-sea flux of heat.

      The pursuit of these three attributes led to examining potential enthalpy referenced to the fixed (surface) pressure $P^0$. The

air-sea heat flux occurs at the sea surface where the pressure is $P^0$, so the air-sea heat flux is the flux of this potential enthalpy,

$h^0$, and hence potential enthalpy possesses the first attribute. Also, since potential enthalpy is a "potential" variable, it

automatically possesses the second attribute.

      As far as the third attribute is concerned, if we first consider mixing processes that are occurring at the sea surface where

the pressure is $P^0$, apart from the heating caused by the dissipation of turbulent kinetic energy $h^0$ is a conservative property at

this pressure, so the third attribute applies to these mixing events. However, for all the turbulent mixing events that take place

deeper in the water column where the pressure exceeds $P^0$ the quantity that is conservative is potential enthalpy referenced to

the pressure of the mixing event, and potential enthalpy referenced to $P^0$ is not conserved. Casting a shadow over potential

enthalpy was the knowledge that enthalpy itself is very sensitive to adiabatic and isohaline changes in pressure (see section

4.1 above) and this would seem to imply (incorrectly) that the pursuit of $h^0$ would not prove to be rewarding. So, the final

hurdle that was needed to show that $h^0$ is an excellent approximation to the heat content per unit mass of seawater was to

quantify the non-conservative production of $h^0$ for turbulent mixing events that occurred at arbitrary pressures in the ocean.

This task was undertaken by McDougall (2003) and Graham and McDougall (2013), and the results are summarised in section





A.18 of the TEOS-10 Manual (IOC et al., 2010) and in section 7 below. In short, it was shown that most of the rather small production of $h^0$ (and of $\Theta$) is due to the dissipation of kinetic energy, $\varepsilon$, and a much smaller part is due to the inherent (or diffusive) non-conservation of $h^0$.

## 7 Thermodynamic Potentials

Many different types of accurate thermodynamic observations (sound speed, freezing point depression, specific heat capacity, etc.) were used by Feistel (2008) to constrain various partial derivatives of what became the TEOS-10 Gibbs function of seawater, and when expressed in terms of enthalpy $h(S_A, T, P)$ and entropy $\eta(S_A, T, P)$ the Gibbs function is

$$g(S_A, T, P) = h(S_A, T, P) - T\,\eta(S_A, T, P). \tag{21}$$

The thermodynamic information contained in enthalpy $h(S_A, T, P)$ is not completely separate to that contained in entropy

$\eta(S_A, T, P)$, but rather, the derivatives of these functions with respect to in-situ temperature must exactly satisfy $h_T = T\eta_T$.

The Gibbs function is unknown and unknowable to the extent

$$[a_1 + a_2 T] + [a_3 + a_4 T]S_A, \tag{22}$$

which means that enthalpy is unknown and unknowable to the extent of $a_1 + a_3 S_A$ and entropy is unknown and unknowable to the extent of $-(a_2 + a_4 S_A)$. No measurement will ever be able to shed light on any of these four coefficients. For example,

terms involving the coefficients $a_3$ and $a_4$ do appear as part of all the terms except $v\mathrm{d}P$ and $P\mathrm{d}v$ in the FTR of Eq. (4), but these terms cancel out of the equation. Because the four coefficients are unknowable, nothing in the measurable universe will ever depend on them. Hence, they are unimportant, as is discussed in several places in the TEOS-10 Manual (IOC et al., 2010). In particular, there is the same amount of oceanographic information in the pair of ocean variables $(S_A, [\eta - a_4 S_A])$ as there is with the $(S_A, \eta)$ pair. Dear reader, if you come across a paper that claims that any of these four coefficients have any

real-world consequences, you are probably reading a paper from one of the for-profit publish-anything journals, and it's best to simply place it in the trash bin where it belongs. Having said that, when the same physical material (e.g. $H_2O$) is being considered in different phases, for example, water in the vapour, liquid and ice phases, the thermodynamic potentials of each phase need to have their arbitrary coefficients made consistent with each other to ensure the equality of the chemical potentials of liquid water, of water vapour and of ice at the triple point. The TEOS-10 thermodynamic potentials of seawater, ice and

humid air have been made consistent with each other in this way so that quantities at the phase boundaries (e.g. freezing temperature and the melting enthalpy) can be accurately calculated (see Feistel et al., 2008).

An alternative thermodynamic potential of seawater, where the temperature variable is Conservative Temperature instead of in-situ temperature, has recently been published in McDougall et al. (2023), namely

$$\hat{\phi}(S_A, \Theta, P) = \hat{h}(S_A, \Theta, P) - c_p^0 \Theta - \int_0^\Theta \hat{\eta}(S_A, \Theta')\mathrm{d}\Theta', \tag{23}$$





and the first two terms, $\hat{h}(S_A, \Theta, P) - c_p^0\Theta$, can also be expressed as the pressure integral $\int_{P_0}^{P} \hat{v}(S_A, \Theta, P')\mathrm{d}P'$ of specific volume

(since we know that $v = \hat{h}_P$ and $c_p^0\Theta \equiv \hat{h}(S_A, \Theta, P_0)$). Both $g(S_A, T, P)$ and $\hat{\phi}(S_A, \Theta, P)$ are thermodynamic potential functions (i. e. "parent" functions) from which all thermodynamic quantities can be evaluated. The Gibbs function has the advantage that its temperature variable, in-situ temperature, is an observable quantity, so that the construction of the Gibbs function from observations of thermodynamic quantities is a manageable (if difficult) task, undertaken in the seminal work of

Feistel (2008). When it comes to using a thermodynamic potential in observational oceanography or in an ocean model, $\hat{\phi}(S_A, \Theta, P)$ has the advantage over the Gibbs function in that its temperature variable, Conservative Temperature, is both a "potential" variable and an almost 100% "conservative" variable.

The thermodynamic potential $\hat{\phi}(S_A, \Theta, P)$ is unique among the known thermodynamic potentials (of which there are now six) in that the thermodynamic information contained in the expressions for enthalpy and for entropy are independent of each

other. When in-situ temperature is used as the temperature variable, the expressions for enthalpy and entropy are not independent of each other but rather their temperature derivatives need to satisfy $(T_0 + t)\eta_T = h_T$. Because of this thermodynamic independence, if both $\hat{h}(S_A, \Theta, P)$ and $\hat{\eta}(S_A, \Theta)$ are known, all the thermodynamic properties of seawater can be calculated (see appendix P of IOC et al., 2010) and the thermodynamic potential, $\hat{\phi}(S_A, \Theta, P)$, is not needed. The TEOS-10 polynomial expressions for $\hat{h}(S_A, \Theta, P)$ and $\hat{\eta}(S_A, \Theta)$ have been published by Roquet et al., (2015) and McDougall et al.

(2023) respectively. McDougall et al. (2023) have also shown that using $\hat{h}(S_A, \Theta, P)$ and $\hat{\eta}(S_A, \Theta)$ to define seawater is considerably more computationally efficient in an ocean modelling context than using the Gibbs function $g(S_A, T, P)$.

Many of the thermodynamic properties that are needed in physical oceanography can be calculated from enthalpy $\hat{h}(S_A, \Theta, P)$ alone without needing the knowledge contained in $\hat{\eta}(S_A, \Theta)$. These variables include internal energy $u$, specific volume $v$, adiabatic compressibility $\kappa$, sound speed $c$, and the conversion between in-situ temperature $t$ and potential

temperature $\theta$ (since $(T_0 + t)/(T_0 + \theta) = \hat{h}_\Theta/c_p^0$). Knowledge of $\hat{\eta}(S_A, \Theta)$ is needed to convert between potential temperature $\theta$ and Conservative Temperature $\Theta$ (since $(T_0 + \theta) = c_p^0/\hat{\eta}_\Theta$) and to calculate the chemical potentials $\mu$ and $\mu^W$ (McDougall et al., 2023).

## 8 Quantifying the non-conservation of several "potential" oceanic variables

Apart from the heating caused by the dissipation of turbulent kinetic energy, enthalpy is conserved when mixing between

fluid parcels occurs at any given pressure, as proven in section 3. Entropy does not have this property, neither does specific volume, nor internal energy, nor total energy, nor potential temperature; none of these properties are conserved when fluid parcels mix at constant pressure. Here we introduce the methods that are used to quantify the extent of the non-conservation of these variables, beginning with specific volume.





### 8.1 Mixing pairs of seawater parcels

Consider the mixing of two seawater parcels with contrasting values of Absolute Salinity and Conservative Temperature. The mixing is assumed to occur to completion, and any dissipation of turbulent kinetic energy is ignored in this analysis and should be considered as a separate issue. Specific volume in taken to be in the functional form $\breve{v}(S_A, h, P)$ where enthalpy $h$ is one of the independent variables. Absolute Salinity and enthalpy are both conserved during the mixing process. Specific volume is expanded in a Taylor series about the final values of Absolute Salinity and enthalpy and the non-conservative

production of specific volume $\delta v$ is found to be (see Graham and McDougall (2013) and IOC et al. (2010))

$$\delta v = -\frac{1}{8}\{\breve{v}_{hh}(\Delta h)^2 + 2\breve{v}_{hS_A}\Delta h \Delta S_A + \breve{v}_{S_A S_A}(\Delta S_A)^2\} \tag{24}$$
$$\approx -\frac{1}{8}\{\hat{v}_{\Theta\Theta}(\Delta\Theta)^2 + 2\hat{v}_{\Theta S_A}\Delta\Theta\Delta S_A + \hat{v}_{S_A S_A}(\Delta S_A)^2\}.$$

The second line of this equation has specific volume expressed in the form $\hat{v}(S_A, \Theta, P)$ and is approximate because of the (very small) non-conservative production of Conservative Temperature during the mixing process: recall that this non-conservation

of $\Theta$ only occurs when the mixing occurs away from the sea surface. Also, these expressions have assumed that the two initial seawater parcels have equal mass: if the masses are unequal, being $m_1$ and $m_1$, with the final mass being $m = m_1 + m_2$, the factor $\frac{1}{8}$ is instead $\frac{1}{2}m_1 m_2/m^2$. Note that this expression (24) is for the mixing between seawater parcels *at a given pressure* and it does not include the thermobaricity process which describes the dianeutral motion as fluid parcels move epineutraly from different pressures until they meet and mix at a given location (see McDougall (1987b), section 11 of this review and

section A22 of IOC et al., 2010).

The corresponding results for the non-conservative production of specific entropy are

$$\delta\eta = -\frac{1}{8}\{\breve{\eta}_{hh}(\Delta h)^2 + 2\breve{\eta}_{hS_A}\Delta h \Delta S_A + \breve{\eta}_{S_A S_A}(\Delta S_A)^2\} \tag{25}$$
$$\approx -\frac{1}{8}\{\hat{\eta}_{\Theta\Theta}(\Delta\Theta)^2 + 2\hat{\eta}_{\Theta S_A}\Delta\Theta\Delta S_A + \hat{\eta}_{S_A S_A}(\Delta S_A)^2\}.$$

Again, the second line here is approximate only because of the very small non-conservation of Conservative Temperature.

In order to calculate the non-conservative production of potential temperature, a similar Taylor series expansion is performed, but in this case of enthalpy in the form $\tilde{h}(S_A, \theta, P)$. When equal masses of two contrasting seawater parcels are mixed the non-conservative production of potential temperature is

$$\delta\theta = \frac{1}{8}(\tilde{h}_\theta)^{-1}\{\tilde{h}_{\theta\theta}(\Delta\theta)^2 + 2\tilde{h}_{\theta S_A}\Delta\theta\Delta S_A + \tilde{h}_{S_A S_A}(\Delta S_A)^2\}. \tag{26}$$

At the sea surface ($P = P_0$) the terms $\tilde{h}_{\theta\theta}$ and $\tilde{h}_{\theta S_A}$ represent the variation of the specific heat capacity $c_p = h_T$ with potential

temperature and Absolute Salinity respectively (see Figure 7).

The calculation for the non-conservative production of Conservative Temperature proceeds similarly, using enthalpy in the form $\hat{h}(S_A, \Theta, P)$, finding

$$\delta\Theta = \frac{1}{8}(\hat{h}_\Theta)^{-1}\{\hat{h}_{\Theta\Theta}(\Delta\Theta)^2 + 2\hat{h}_{\Theta S_A}\Delta\Theta\Delta S_A + \hat{h}_{S_A S_A}(\Delta S_A)^2\}. \tag{27}$$




### 8.2 The causes of the non-conservation of several variables

We begin with this last expression, (27), and concentrate on the second order derivatives of $\hat{h}(S_A, \Theta, P)$. The first and second $\Theta$ derivates of $\hat{h}(S_A, \Theta, P)$ are

$$\hat{h}_\Theta \;=\; c_p^0 + \int_{P_0}^{P} \hat{v}_\Theta\big(S_A, \Theta, P'\big)\mathrm{d}P', \qquad \text{and} \qquad \hat{h}_{\Theta\Theta} \;=\; \int_{P_0}^{P} \hat{v}_{\Theta\Theta}(S_A, \Theta, P')\mathrm{d}P', \tag{28}$$

and corresponding expressions exist for $\hat{h}_{\Theta S_A}$ in terms of the pressure integral of $\hat{v}_{\Theta S_A}$, and for $\hat{h}_{S_A S_A}$ in terms of the pressure integral of $\hat{v}_{S_A S_A}$. These expressions for the second derivatives of $\hat{h}(S_A, \Theta, P)$ show that the non-conservative production of

Conservative Temperature, Eq. (24), is due to the nonlinear nature of $\hat{h}(S_A, \Theta, P)$ (or equivalently of $\hat{v}(S_A, \Theta, P)$), as a function of $S_A$ and $\Theta$. That is, the non-conservative production of $\Theta$ is due only to the nonlinear nature of $\hat{h}(S_A, \Theta, P)$ and is independent of the nonlinear nature of $\hat{\eta}(S_A, \Theta)$.

Now considering the expression (26) for the non-conservative production of potential temperature, the first and second $\theta$ derivates of $\tilde{h}(S_A, \theta, P)$ are

$$\tilde{h}_\theta \;=\; \tilde{h}_\theta^0(S_A, \theta) + \int_{P_0}^{P} \tilde{v}_\theta\big(S_A, \theta, P'\big)\mathrm{d}P', \qquad \text{and} \qquad \tilde{h}_{\theta\theta} \;=\; \tilde{h}_{\theta\theta}^0(S_A, \theta) + \int_{P_0}^{P} \tilde{v}_{\theta\theta}(S_A, \theta, P')\mathrm{d}P'. \tag{29}$$

The terms in the pressure integrals here are very similar in magnitude to those in Eq. (28), but the difference between the expressions for $\tilde{h}_{\theta\theta}$ and $\hat{h}_{\Theta\Theta}$ is the additional term $\tilde{h}_{\theta\theta}^0(S_A, \theta)$ in Eq. (29), where the superscript 0 denotes that the property is evaluate at the sea surface, that is, at pressure $P_0$. The term in the pressure integral of $\tilde{v}_{\theta\theta}$ in Eq. (29) is zero at the sea surface and increases in magnitude with pressure becoming comparable with the first term, $\tilde{h}_{\theta\theta}^0(S_A, \theta)$, at $\sim 10^8\,\mathrm{Pa} =$

10,000 dbar. However, at these depths the temperature and salinity changes in the ocean are tiny compared with those in the upper ocean so the non-conservative production of potential temperature, $\delta\theta$, is very small. In order to find the root cause of the terms $\tilde{h}_{\theta\theta}^0(S_A, \theta)$, $\tilde{h}_{\theta S_A}^0(S_A, \theta)$ and $\tilde{h}_{S_A S_A}^0(S_A, \theta)$ that cause the non-conservative production of potential temperature at $P_0$, the relationship $(T_0 + \theta) = c_p^0/\hat{\eta}_\Theta$ is used to write $\tilde{h}_\theta^0(S_A, \theta)$ as $-(\hat{\eta}_\Theta)^2/\hat{\eta}_{\Theta\Theta}$ and $\tilde{h}_{\theta\theta}^0(S_A, \theta)$ is $\tfrac{1}{2}(c_p^0)^{-1}\big(\big[(\hat{\eta}_\Theta)^2/\hat{\eta}_{\Theta\Theta}\big]^2\big)_\Theta$. This shows that $\tilde{h}_{\theta\theta}^0(S_A, \theta)$ depends on $\hat{\eta}(S_A, \Theta)$ and is independent of $\hat{h}(S_A, \Theta, P)$. Making use of several results from Appendix

P of IOC et al. (2010), it can be shown that the same conclusion applies to the terms $\tilde{h}_{\theta S_A}^0(S_A, \theta)$ and $\tilde{h}_{S_A S_A}^0(S_A, \theta)$ in Eq. (26). That is, these terms are also due to the nonlinear nature of $\hat{\eta}(S_A, \Theta)$ and are independent of the nonlinearity of $\hat{h}(S_A, \Theta, P)$. We conclude that of our expressions $\hat{h}(S_A, \Theta, P)$ and $\hat{\eta}(S_A, \Theta)$ that serve to define the thermodynamic properties of seawater, the dominant terms (the terms that apply at $P_0$) that cause potential temperature to not be conserved on mixing are due to the nonlinearity in entropy $\hat{\eta}(S_A, \Theta)$ and not due to the nonlinearity in enthalpy $\hat{h}(S_A, \Theta, P)$. The nonlinear nature of enthalpy

plays a much smaller role and only does so deeper in the water column.

Exactly the same conclusion applies to entropy as can be seen from Eq. (25). When mixing occurs at the sea surface, $\Theta$ is a conservative variable and the second line of Eq. (25) implicates only $\hat{\eta}(S_A, \Theta)$ in the non-conservative production of entropy. For mixing deeper in the ocean, Conservative Temperature is not totally conserved (due to the nonlinear nature of $\hat{h}(S_A, \Theta, P)$,





not $\hat{\eta}(S_A, \Theta)$) and this non-conservation adds to the non-conservation of entropy caused by $\hat{\eta}(S_A, \Theta)$. Note that the non-
conservation of $\Theta$ is sign-definite and positive (though small) and adds to the non-conservative production of entropy (Graham
and McDougall, 2013). It is interesting that the non-conservative nature of both $\theta$ and $\eta$ is caused by the same nonlinear nature
of entropy $\hat{\eta}(S_A, \Theta)$, with further non-conservation due to $\hat{h}(S_A, \Theta, P)$ that applies away from the sea surface and is smaller by
two or three orders of magnitude (see the next subsection and Figure 6). Note that the sign-definite nature of the non-
conservative production of $\Theta$ is not a consequence of the Second Law of Thermodynamics.

When expressed in terms of constraints on the Gibbs function $g(S_A, T, P)$, it is well known from studying the instantaneous
evolution equations that the Second Law of Thermodynamics requires that $g_{TT} < 0$ and $g_{S_A S_A} > 0$. The physical
interpretations of these constraints are that when a seawater parcel is heated, its temperature must increase, and that salt must
diffuse down the gradient of salinity, not up this gradient. Appendix A16 of the TEOS-10 Manual (IOC et al., 2010) started
from the conservative nature of enthalpy when parcels are mixed at constant pressure and proved that exactly the same pair of
constraints apply in the presence of turbulent mixing processes.

The non-conservation of specific volume, like that of Conservative Temperature, is caused only by the nonlinear nature of
enthalpy $\hat{h}(S_A, \Theta, P)$ and is not affected by the non-linear nature of entropy $\hat{\eta}(S_A, \Theta)$ This can be found by examining Eq. (24)
where the terms $\hat{v}_{\Theta\Theta}$, $\hat{v}_{\Theta S_A}$ and $\hat{v}_{S_A S_A}$ can be written as $\hat{h}_{P\Theta\Theta}$, $\hat{h}_{P\Theta S_A}$ and $\hat{h}_{P S_A S_A}$, and the extra production of specific
volume that occurs in the first line of (24) compared with the second line, is due to the non-conservative production of
Conservative Temperature which, as we have noted below Eq. (28), is also due to the nonlinear nature of $\hat{h}(S_A, \Theta, P)$ and
not of $\hat{\eta}(S_A, \Theta)$. It is also interesting to note that similar form of the dependence of the non-conservation of $v$ and $\Theta$ on
the second derivatives of specific volume. That is, the same $\hat{v}_{\Theta\Theta}$, $\hat{v}_{\Theta S_A}$ and $\hat{v}_{S_A S_A}$ terms appear in both Eqs. (24) and (27,
28); with the latter case being in the form of pressure integrals.

In conclusion, in this subsection we have shown that the non-conservative natures of both Conservative Temperature
$\Theta$ and specific volume $v$ are caused by the nonlinear nature of enthalpy $\hat{h}(S_A, \Theta, P)$ and are independent of the non-
linearity of entropy $\hat{\eta}(S_A, \Theta)$. In contrast, the non-conservative natures of both potential temperature and entropy are
predominantly due to the nonlinear nature of entropy $\hat{\eta}(S_A, \Theta)$, with only a very minor role, at large pressures, for the
nonlinear nature of enthalpy $\hat{h}(S_A, \Theta, P)$.

Before closing this subsection, we discuss the non-conservative production of internal energy $u$ and total energy $E \equiv u +$
$0.5\mathbf{u} \cdot \mathbf{u} + \Phi$. Figure 8 shows a water column on the left before a mixing even occurs between two shaded vertically adjacent
seawater parcels with contrasting Conservative Temperature and Absolute Salinity, and on the right the water column is shown
after the mixing event. The mixing process takes place at pressure $P^m$. Because of the non-conservative behaviour of specific
volume, the volume of the final mixed fluid is less than the sum of the two initial volumes and all the water parcels in the water
column above the mixing event retain their pressure, internal energy, enthalpy, but suffer a loss of height and of gravitational
potential energy. All these parcels experience a change in their total energy even though they have not experienced a mixing





event. This occurs because total energy $E \equiv u + 0.5\mathbf{u} \cdot \mathbf{u} + \Phi$ is not a thermodynamic quantity, that is, it depends not only on $(S_A, \Theta, P)$ but also on the extra quantities, kinetic energy and gravitational potential energy, which are not governed by thermodynamics.

Concentrating now on the fluid that undergoes the mixing in Figure 8, if we ignore any changes in kinetic energy, the non-
conservative production of total energy, $\delta E$, will be the same as that of internal energy, $\delta u$. Since enthalpy and pressure are unchanged during the mixing process ($\delta h = 0$ and $P = P^m$), the definition of enthalpy as $h = u + Pv$ is used to deduce that $\delta u = -P^m \delta v$. When examining Eqs. (24), (27) and (28) we find an approximate relationship between the non-conservative production of $v$ and of $\Theta$, namely $c_p^0 \delta \Theta \cong -(P^m - P_0)\delta v \cong -P^m \delta v = \delta u = \delta E$. This approximation relies on the second derivative terms such as $\hat{v}_{\Theta\Theta}$ in Eq. (28) not being strong functions of pressure.

**8.3 Comparing the non-conservation of several variables**

The relationships (24) – (27) are illustrated in Figures 2–5; these figures are explained in more detail in sections A16-A19 of the TEOS-10 Manual (IOC et al., 2010). The variable that is contoured in Figure 2 was formed by first subtracting from specific volume the linear function of Absolute Salinity that made the result zero at the two $(S_A, \Theta)$ locations $(0 \text{ g kg}^{-1}, 0°C)$ and $(35.16504 \text{ g kg}^{-1}, 0°C)$, second by scaling the result to be equal to $25°C$ at $(35.16504 \text{ g kg}^{-1}, 25°C)$ and third,
subtracting Conservative Temperature $\Theta$. If specific volume at 0 dbar were a linear function of Absolute Salinity and Conservative Temperature, Figure 2 would be populated with zeros. Instead, what is contoured in this figure enables us to evaluate the non-linearity of specific volume on the $(S_A, \Theta)$ diagram, expressed in °C. Specifically, the contoured variable indicates what warming or cooling would be required to account for the difference between the actual specific volume and a linear equation of state version of specific volume, using the thermal expansion coefficient applicable at $S_A = $
$35.16504 \text{ g kg}^{-1}$, $\Theta \approx 12.5°C$ and $p = 0$ dbar. The main use of these Figures 3 – 6 is in estimating the relative magnitudes of the non-conservative production/destruction of various thermodynamic variables. As an example, consider the mixing of equal masses of the seawater parcels at $(0 \text{ g kg}^{-1}, 0°C)$ and $(35.16504 \text{ g kg}^{-1}, 25°C)$. For Figure 2 we see that the mixture, at the mid-point Absolute Salinity and Conservative Temperature, would require warming by approximately 7.5°C in order for its specific volume to be the average of the specific volumes of the two original seawater parcels.

Figure 3 is the corresponding plot for entropy, with the variable that is contoured on Figure 3 being specific entropy multiplied by a dimensional constant so that the result is $25°C$ at $(35.16504 \text{ g kg}^{-1}, 25°C)$ and then Conservative Temperature $\Theta$ is subtracted. When the same two seawater parcels, $(0 \text{ g kg}^{-1}, 0°C)$ and $(35.16504 \text{ g kg}^{-1}, 25°C)$, are mixed in equal proportions, Figure 3 shows that the entropy produced is the same as would be produced by approximately 0.45°C of warming.

Figure 4 is the corresponding plot for potential temperature, with the contours being the difference between potential
temperature and Conservative Temperature, $\theta - \Theta$, at $p = 0$ dbar. When the same two seawater parcels, $(0 \text{ g kg}^{-1}, 0°C)$ and $(35.16504 \text{ g kg}^{-1}, 25°C)$, are mixed in equal proportions, Figure 4 shows that the potential temperature of the mixture is less than its Conservative Temperature by approximately 0.35°C. In this case the non-conservative behaviour of potential





temperature has acted to destroy (rather than produce) potential temperature. If potential temperature were carried as the model's temperature variable in an ocean model, the potential temperature would be conserved during this mixing process and

so the potential temperature would be overestimated by 0.35℃. In this way the entropy produced during the mixing process would be too large by 78% (being 0.35/0.45 of 100%, with the 0.45℃ figure coming from the previous paragraph, representing the actual production of entropy). This 78% overestimation of the non-conservation production of entropy would occur for any mixing process occurring along the line joining $(0\,\mathrm{g\,kg^{-1}}, 0℃)$ and $(35.16504\,\mathrm{g\,kg^{-1}}, 25℃)$. The curvature of the isolines on Figure 4 changes sign at the higher salinities and in this region of $(S_A, \Theta)$ space the non-conservative production of

potential temperature is positive.

     While it is possible to consider the non-conservative production when pairs of parcels mix on Figure 4, how can we obtain a realistic measure of the total effects of such non-conservation as reflected in the present ocean state. That is, we cannot sum up all the unknowable number of individual non-conservation events that have occurred in the past thousand years in all of the ocean. It turns out that the contoured variable of Figure 4, $\theta - \Theta$, is the error that is made by interpreting an ocean model's

temperature variable as potential temperature. This is literally the case for an ocean model that is driven by imposed air-sea heat fluxes, and the same error measure approximately applies when the air-sea boundary condition is a combination of restoring and flux conditions. This error is easily avoided by simply interpreting the ocean model's temperature variable as Conservative Temperature (McDougall et al. (2021a) and the following section of this review).

     We come now to the non-conservative production of Conservative Temperature illustrated in Figure 5. Since Conservative

Temperature is a conservative variable for mixing at the sea surface, we illustrate its non-conservation at a pressure of 600dbar where its non-conservation is maximised (McDougall, 2003). Enthalpy evaluated at 600dbar is a conservative quantity for mixing processes at this pressure and is used as the vertical axis of Figure 5. When again considering the mixing of equal masses of the seawater parcels at $(0\,\mathrm{g\,kg^{-1}}, 0℃)$ and $(35.16504\,\mathrm{g\,kg^{-1}}, 25℃)$ (two of the bold black dots on Figure 5) we see that the non-conservative production of Conservative Temperature amounts to 0.002℃ (2 mK).

On the basis of the above examination of mixing between the seawater parcels $(0\,\mathrm{g\,kg^{-1}}, 0℃)$ and $(35.16504\,\mathrm{g\,kg^{-1}}, 25℃)$, the non-conservative production of the three variables $\eta$, $\theta$ and $\Theta$, the relative importance of the non-conservation of these variables in the ocean would seem to be in the ratio $450: 350: 2$, or $250: 175: 1$. On the other hand, simply looking at the maximum contoured values in Figures 4, 5 and 6 one might guess that the relevant ratio of the relative extent of the non-conservation of these variables is $0.5: 1.8: 0.003$, or $167: 600: 1$. However, neither of these estimates

considers the ranges of salinity and temperature over which the interior mixing processes occur in the ocean. To do this Graham and McDougall (2013) evaluated these non-conservative production terms in an ocean model. The first step in doing this was to develop the evolution equations for these variables in a turbulent ocean, and this was based on the knowledge that enthalpy is conserved when mixing proceeds between fluid parcel at a given pressure, as we have discussed in section 4.2 above.

Graham and McDougall (2013) began with our Eq. (15) above which applies instantaneously prior to averaging over unresolved turbulent motions. This equation was then averaged over turbulent motions including lateral mesoscale motions,





which are then parametrized with an epineutral scalar diffusivity $K$, and small-scale isotropic turbulence which is parameterised with an isotropic small-scale turbulent diffusivity $D$. The resulting evolution equation for potential enthalpy referenced to the pressure of the mixing processes, in Boussinesq form, is

$$\mathrm{d}h^m/\mathrm{d}t \;=\; \gamma_z \nabla_n \cdot (\gamma_z^{-1} K \nabla_n h^m) + (D h_z^m)_z + \varepsilon, \qquad\qquad \text{at } P = P^m \qquad (30)$$

where the total derivative, $\mathrm{d}h^m/\mathrm{d}t$, is with respect to the Temporal-Residual Mean velocity of McDougall and McIntosh (2001), and $h^m$ here is the thickness-weighted value, having been averaged between closely spaced neutral density ($\gamma$) surfaces; also, in the following equation $\Theta$ and $S_A$ are also the thickness-weighted values. Graham and McDougall then exploited the fact that at the pressure $P^m$ of a particular mixing event, the potential enthalpy $h^m = \hat{h}(S_A, \Theta, P^m)$ is a function

only of Absolute Salinity and Conservative Temperature, arriving at the evolution equation for $\Theta$ in a turbulent ocean,

$$
\begin{aligned}
\mathrm{d}\Theta/\mathrm{d}t \;=\;& \gamma_z \nabla_n \cdot (\gamma_z^{-1} K \nabla_n \Theta) + (D\Theta_z)_z + \varepsilon/\hat{h}_\Theta \\
& + K(\hat{h}_\Theta)^{-1}(\hat{h}_{\Theta\Theta} \nabla_n \Theta \cdot \nabla_n \Theta + 2\hat{h}_{\Theta S_A} \nabla_n \Theta \cdot \nabla_n S_A + \hat{h}_{S_A S_A} \nabla_n S_A \cdot \nabla_n S_A) \\
& + D(\hat{h}_\Theta)^{-1}(\hat{h}_{\Theta\Theta} \Theta_z^2 + 2\hat{h}_{\Theta S_A} \Theta_z S_{A_z} + \hat{h}_{S_A S_A} S_{A_z} S_{A_z}). \qquad\qquad \text{at } P = P^m \qquad (31)
\end{aligned}
$$

The same approach was used to develop the evolution equations for potential temperature and for specific entropy. In each

case the non-conservative production terms in the evolution equation have terms that are immediately recognisable from the corresponding terms in the two-parcel mixing expressions (25), (26) and (27).

Graham and McDougall (2013) vertically integrated the non-conservative production terms in these evolution equations for $\Theta$, $\theta$ and $\eta$ finding values of the air-sea heat flux, as a function of longitude and latitude, that would be required to balance the depth-integrated non-conservative source terms. Figure 6 shows the results as a histogram of these equivalent air-sea heat

fluxes. The relative magnitudes of the non-conservation of these variables can be gauged from the values greater than which only 5% of the values occur; these values are $1200 \text{ mW m}^{-2}$ ($1.2 \text{ W m}^{-2}$) for specific entropy $\eta$, $120 \text{ mW m}^{-2}$ for potential temperature $\theta$, and $1 \text{ mW m}^{-2}$ for Conservative Temperature $\Theta$. That is, the relative amounts by which $\eta$, $\theta$ and $\Theta$ and not conservative in the ocean are $1200:120:1$ rather than the previously mentioned values of these ratios. In addition, Graham and McDougall (2013) found that the *mean* effective air-sea flux due to the non-conservation of Conservative Temperature is

approximately $0.3 \text{ mW m}^{-2}$, and that if the air-sea flux arrives into the ocean deeper by 25m than the depth at which most of this heat flux leaves the ocean, this leads to a mis-estimation of the net air-sea heat flux of $0.6 \text{ mW m}^{-2}$ (McDougall et al., 2021a). These values can be compared with the *mean* value of the depth-integrated dissipation of turbulent kinetic energy which is approximately $10 \text{ mW m}^{-2}$, demonstrating that the non-conservation of Conservative Temperature less important by a factor of at least ten than the neglect of the dissipation of turbulent kinetic energy as a source of ocean warming.

The five steps that have led to these estimates of the non-conservation of various thermodynamic variables are

    1.   realizing that enthalpy is a conservative quantity when mixing occurs at fixed pressure,



2.  noting that potential enthalpy $h^m = \hat{h}(S_A, \Theta, P^m)$ referenced to the pressure of a mixing event, $P^m$, is also conserved during the mixing event, and has the advantage over enthalpy itself in that $h^m$ is unchanged during the variations in pressure that predate the mixing event,

3.  deriving the evolution equation for $h^m = \hat{h}(S_A, \Theta, P^m)$ in a turbulent ocean by carefully averaging the First Law of Thermodynamics,

4.  using this equation to find evolution equations for Conservative Temperature, potential temperature and entropy, which include the relevant non-conservative production terms, and

5.  evaluating the magnitudes of the non-conservative terms in these evolution equations.

Having concentrated above on the typical, or root-mean-square values of the non-conservation of various oceanographic variables, we also mention that an extreme value of $\theta - \Theta$ of $-1.5°C$ occurs when warm fresh river water (e.g. the Amazon) flows into the ocean. That is, the enthalpy per unit mass of this river water is given by its transport of Conservative Temperature which is 1.5°C greater than potential temperature for this warm fresh water.

## 9 The temperature and salinity variables of ocean and climate models

This section will show that because ocean models to date have not included non-conservative source terms in their evolution equations for either temperature or salinity, the model's temperature and salinity must be interpreted as Conservative Temperature $\Theta$ and Preformed Salinity $S_*$. It is observation-derived fields of $\Theta$ and $S_*$ that should be used to both initialize these ocean models and to compare with model output. During the running of an ocean model, if the air-sea heat flux is parameterized using a bulk formula based on potential temperature $\theta$ at the sea surface (SST), then it needs to be calculated at

run time using the relationship $(T_0 + \theta) = c_p^0 / \hat{\eta}_\Theta$ in terms entropy in the functional form $\hat{\eta}(S_A, \Theta)$ (McDougall et al., 2023).

The following subsection 9.1 discusses the correct interpretation of model salinity and shows that the neglect to date of estimating Absolute Salinity in ocean models has led all these models to misestimate the meridional overturning transport of key water masses by ~13.5%. Section 9.2 below goes on to show that the temperature variable in ocean models should be interpreted as being Conservative Temperature, and if one insists that the model temperature is potential temperature, then up

to 0.3 W m$^{-2}$ of the air-sea heat flux is lost in some regions; that is, up to 0.3 W m$^{-2}$ leaves the atmosphere but does not arrive in the ocean. Such non-conservative spontaneous heat production/loss can no longer be countenanced in climate models.

### 9.1 The salinity of ocean models is Preformed Salinity $S_*$

The influence of biogeochemistry on salinity, specific volume and the thermal wind equation has been ignored in ocean modelling to date. Biogeochemistry causes spatial variations in the relative concentrations of the constituents of sea salt

(particularly variations in silicic acid concentration), and these spatial differences have been ignored and so have not been allowed to affect the model's salinity and specific volume. If an ocean model carried several biogeochemical variables, then the Absolute Salinity $S_A$ could be calculated as an addition to the model's Preformed Salinity $S_*$ using the following equation,



$$(S_A - S_*)/(\mathrm{g\,kg^{-1}}) \; = \; (73.7\,\Delta\mathrm{TA} \; + \; 11.8\,\Delta\mathrm{DIC} \; + \; 81.9\,\mathrm{NO_3^-} \; + \; 50.6\,\mathrm{Si(OH)_4})/(\mathrm{mol\,kg^{-1}})\,, \qquad (32)$$

which was developed by Pawlowicz *et al.* (2011); see the discussion below Eq. (3) above for a description of the variables in this equation. In this approach, the ocean model continues to carry Preformed Salinity $S_*$ as one of its prognostic variables and evaluates Absolute Salinity $S_A$ using Eq. (32) as a prelude before every call to the equation of state. This method has not yet been implemented by an ocean model. An alternative method of allowing for the variations of seawater composition by relaxing towards existing observations of these quantities has been outlined in IOC et al. (2010), Wright et al. (2011), McDougall and Barker (2011) and McDougall et al. (2013), but this method has also not been implemented to date.

Ocean models initialize their salinity as Reference Salinity (or, in the EOS-80 case, as Practical Salinity). At the sea surface the concentration of nutrients and silicic acid is very small so that Reference Salinity at the sea surface is virtually the same as Preformed Salinity (and to Absolute Salinity). Since ocean models are initialized (and usually restored) to surface values of Preformed Salinity, and since both the models' salinity and Preformed Salinity are conservative, then the output salinity of models has only one interpretation, named Preformed Salinity (or $S_*/u_{\mathrm{PS}}$ in the case of an EOS-80 based ocean model, see Eq. (1)).

How large might be the influence of neglecting $(S_A - S_*)$ in ocean models? From section A5 and A20 of the TEOS-10 Manual (IOC et al. (2010)) we see that of the ocean that is deeper than 1000 dbar in the World Ocean, 58% of the locations would misestimate the thermal wind balance by ~2% due to ignoring the difference between Absolute and Reference Salinities. The corresponding misestimation in an ocean model context, namely the effect on the thermal wind balance arising from $(S_A - S_*)$ is 2.7%, being larger than 2% by a factor of 1.35 (Figure 1).

McCarthy et al. (2015) studied the influence of using Absolute Salinity versus Reference Salinity in calculating the overturning circulation in the North Atlantic. They found that the overturning streamfunction changed by 0.7Sv at a depth of 2700m, relative to a mean value at this depth of about 7 Sv, i.e., a 10% effect. Since the salinity variable in ocean models is actually Preformed Salinity, the neglect of the distinction between Preformed and Absolute Salinities in ocean models means that they currently misestimate the overturning streamfunction at this depth by 1.35 (see Figure 1) times 0.7Sv, namely ~1Sv, which is 13.5% of the overturning streamfunction. It surely is high time to include a scheme in ocean models to account for the difference between Absolute Salinity and the model's Preformed Salinity in order to avoid these transport errors of 13.5%.

As a final remark on salinity in ocean models, we note that ocean models that use the EOS-80 equation of state usually describe their salinity variable as being Practical Salinity. This is not the case. These models do not include the non-conservative term that appears in the evolution equation for Practical Salinity, and so the correct interpretation of the salinity variable in these models is $S_*/u_{\mathrm{PS}}$). These issues are discussed in greater detail in McDougall et al. (2021a).

We conclude that because ocean models, of both the TEOS-10 and EOS-80 variety, treat their salinity variable as being conservative, the salinity variable is neither Absolute Salinity nor Reference Salinity (nor Practical Salinity in the case of EOS-80 models) but rather is Preformed Salinity $S_*$ (or $S_*/u_{\mathrm{PS}}$ in the case of EOS-80 models). The neglect of any attempt in ocean



models to enable the evaluation of specific volume using Absolute Salinity means that overturning transports are currently in error by an estimated 13.5%.

## 9.2 The temperature of ocean models is Conservative Temperature $\Theta$

The air-sea heat flux in ocean models is related to the air-sea flux of the model's temperature variable using a fixed value of the specific heat $c_p^0$. This is appropriate when the model temperature is interpreted as being Conservative Temperature as it is

in TEOS-10 based ocean models. But what are the implications for an EOS-80 based ocean model where the model temperature is taken to be potential temperature $\theta$? This question is examined in detail in McDougall et al. (2021a). In this situation the specific heat capacity that should be used to relate the air-sea heat flux to the flux of potential temperature is $c_p(S_*, \theta, P_0)$, the specific heat capacity at the sea surface pressure and at the sea surface Preformed Salinity and sea surface temperature $\theta$. With the air-sea heat flux being $Q$, the error in using the fixed specific heat capacity $c_p^0$ is

$$\Delta Q \ = \ Q\big[c_p(S_*, \theta, P_0)/c_p^0 - 1\big] \ = \ Q\big[1/\hat{\theta}_\Theta - 1\big]. \tag{33}$$

McDougall et al. (2021a) showed that this difference between the heat flux leaving the atmosphere compared to that entering the ocean is greater than $0.3 \text{ W m}^{-2}$ in some regions of the ocean (see Figure 5c of McDougall et al., 2021a), and $2\frac{1}{2}\%$ of the locations in the world ocean have a $\Delta Q > 0.135 \text{ W m}^{-2}$. It is disquieting to have the ocean lose this heat flux which the atmosphere thinks it is exchanging, especially when considered in relation to the net air-sea heat flux due to anthropogenic

warming over the past several decades of $0.3 \text{ W m}^{-2}$ (Zanna et al. 2019).

When this same analysis is performed in a climate that is 2C warmer, the missing surface heat flux when one insists on interpreting the model's temperature variable as potential temperature $\theta$ increases by 10% (Bob Hallberg and Ryan Holmes, personal communication, 2023). Hence, even when comparing a climate change run with a control run, 10% of the effects in the coupled system of the missing surface heat flux remains. This 10% increase is readily explained with reference to Eq. (33)

and Figure 7:- when the range of sea surface temperatures increases by 10% (from the fixed freezing temperature to the 10% warmer surface temperatures, on average), so too does the spatial variation of the specific heat capacity.

Fortunately, there is a very simple and effective solution to this issue, namely, to interpret the temperature variable in EOS-80 based ocean models as being Conservative Temperature $\Theta$. While the equation of state in an EOS-80 based ocean model expects to be called with potential temperature, McDougall et al. (2021a) have shown that the differences in the horizontal

density gradient and the thermal wind equation caused by a switch from potential temperature to Conservative Temperature is small compared with the disappearing heat fluxes at the sea surface and also compared with how accurately the expression for specific volume and the thermal expansion coefficient is known from the original laboratory measurements of the 1950s and 1960s.

One could attempt to retain potential temperature as the temperature variable of an EOS-80 based ocean model, but doing

so encounters the following five issues.

1.    It is not possible to accurately choose the value of the isobaric heat capacity at the sea surface that is needed when $\theta$ is





the model's temperature variable. The problem arises because of unresolved spatial and temporal variations in the sea surface salinity (SSS) and SST [for example, unresolved rain events that temporarily lower the SSS but are not represented in the time-averaged data]. These unresolved variations in SSS and SST act in conjunction with the nonlinear dependence of the isobaric specific heat on salinity and temperature to mean that it is not possible to obtain the appropriately averaged value of the isobaric specific heat.

2. It is not possible to accurately estimate the non-conservative source terms for $\theta$. These terms are the product of a turbulent flux and a mean gradient, and in an eddy-resolved ocean model, how would one go about finding the eddy flux of $\theta$, which depends on how the averaging is done in space and time. There are issues here about how to calculate the appropriate mean gradients, over what space and time scales, and how to treat non-divergent eddy fluxes.

3. Calculating the meridional heat flux through an ocean section cannot be done accurately if $\theta$ is the model's temperature variable, because of the interior source terms.

4. Because of having an inconsistent air-sea heat flux and only approximate estimates of the non-conservative source terms, the interior potential temperature would have errors that would cause errors in the horizontal density gradient and so in the thermal wind (vertical shear).

5. Ocean modelers often use the conservation of salinity and the model's temperature variable to check the model's numerical consistency. If $\theta$ is adopted as the model variable, this is no longer possible because $\theta$ is not a conservative variable.

These five issues are easily avoided by simply interpreting the model temperature variable as being Conservative Temperature $\Theta$ in not only TEOS-10 ocean models, but also in EOS-80 based models.

In addition to salinity and temperature, many authors have written about the desirability of a variable whose isolines are in some sense "perpendicular" to potential density on the $S_A - \Theta$ diagram. McDougall et al. (2021b) reviewed these different suggested versions of such "spice" variables and concluded that those that depend of the isolines of spice and potential density being normal to each other on the SA-CT diagram depend on the relative scales that are chosen for the axes of salinity and temperature on the SA-CT diagram, and do not accurately represent the contrast in water mass properties along isopycnals. In contrast a "spice" variable that makes some physical sense is that of McDougall and Jackett (1985) and McDougall and Krzysik (2015) whose variations along isopycnals represents the propensity for double-diffusive convection, that is, the compensating influence of salinity and temperature on density.

## 10 The neutral tangent plane

The idea that lateral mixing in the ocean occurs predominantly along some type of isopycnal surface dates back to Iselin (1939) but has only been physically justified relatively recently by the following rather simple argument published in section 7.2 of Griffies (2004), section 2 of McDougall and Jackett (2005a) and in more detail in section 1 of McDougall et al. (2014). The argument is actually a *counter argument* (illustrated in Figure 9) where one imagines that lateral mixing *does not* occur along the neutral tangent plane, so that as a seawater parcel is moved a small distance laterally it finds itself either above or below the locally referenced potential density surface. If this counter argument were true the seawater parcel would then have a




different in-situ density to that of the surrounding ocean at this location, leading to the parcel being accelerated vertically by the Archimedean buoyant force. This vertical motion would create a vertical plume of turbulent motion that would result in turbulent diapycnal mixing. The ocean observation that rules out this behaviour of is the rather small diapycnal mixing in the ocean interior. The interior diapycnal mixing that is observed is well explained by the expected breaking of internal gravity waves and lee waves, so that there is no room in the observations of interior diapycnal mixing to accommodate extra diapycnal mixing arising from the possibility of lateral mixing occurring in a non-neutral manner.

In fluid dynamics turbulent mixing is envisaged to occur in a two-stage process. First there is an advection stage where fluid parcels are moved (or exchanged) in an adiabatic and isentropic manner followed by a second stage where diapycnal mixing (and intimate molecular diffusion) occurs. The same two-stage approach applies to lateral mixing along the neutral tangent plane. In the first stage a fluid parcel is moved to a new location where the infinitesimal change of pressure is $\delta P$ while the Absolute Salinity and Conservative Temperature of this seawater parcel are unchanged (since these properties are both "potential" properties) so that the parcel's in-situ density has changed by $\rho\kappa\delta P$ where $\kappa$ is the adiabatic and isentropic compressibility. At this new location the ocean environment that surrounds the isolated parcel has an Absolute Salinity that is $\delta S_A$ different to that at the original location and a Conservative Temperature that is different by $\delta\Theta$. The in-situ density of the surrounding ocean at this new location is different to that at the original location by $\rho(\kappa\delta P + \beta\delta S_A - \alpha\delta\Theta)$ where $\beta$ and $\alpha$ are the saline contraction coefficient and thermal expansion coefficient respectively. As discussed in the previous paragraph, the key property of the neutral tangent plane is that seawater parcels do not experience Archimedean buoyant forces when moved small distances in a neutral direction and this requires that the in-situ density of our isolated seawater parcel is the same as the in-situ density of the surrounding ocean at this location, that is, $\rho\kappa\delta P = \rho(\kappa\delta P + \beta\delta S_A - \alpha\delta\Theta)$. Hence along a neutral trajectory in the ocean the variations of Absolute Salinity and Conservative Temperature must obey $\beta\delta S_A - \alpha\delta\Theta$ which applies to both spatial and temporal changes, so that in the limit we have the differential relationships (McDougall, 1987a)

$$\alpha\nabla_n\Theta = \beta\nabla_n S_A, \qquad \text{and} \qquad \alpha\frac{\partial\Theta}{\partial t}\Big|_n = \beta\frac{\partial S_A}{\partial t}\Big|_n, \tag{34}$$

where $\nabla_n$ denotes the spatial gradient operator in the neutral tangent plane.

If the seawater parcel described above continues to not mix with its surroundings but instead retains its original Absolute Salinity and Conservative Temperature as it moves finite distances around the ocean (as a sub-mesoscale coherent vortex, SCV) the property of "zero Archimedean buoyant force" for such a finite-amplitude lateral excursion is described by a zero value of the specific volume anomaly $\tilde\delta$ defined by

$$\tilde\delta(S_A, \Theta, P) \equiv v(S_A, \Theta, P) - v(\tilde S_A, \tilde\Theta, P). \tag{35}$$

Such a seawater parcel with the constant values $\tilde S_A$ and $\tilde\Theta$ moves on the ocean's $\tilde\delta = 0$ surface, and this "SCV" behaviour has been discussed and quantified by McDougall (1987c) and Lang et al. (2020).

The epineutral gradient of $\tilde\delta$ in the neutral tangent plane can be shown to be (from Klocker et al. (2009) and McDougall and Klocker (2010))





$$\tilde{\rho}^{\Theta}\nabla_n\tilde{\delta} \approx T_b(\Theta - \widetilde{\Theta})\nabla_n P \tag{36}$$

where $\tilde{\rho}^{\Theta}$ is potential density referenced to the pressure $\tilde{P}$ and $T_b$ is the thermobaric parameter $T_b \equiv \hat{\alpha}_P - (\hat{\alpha}/\hat{\beta})\hat{\beta}_P$ where the over-hats indicate that these variables are functions of $(S_A, \Theta, P)$. The corresponding epineutral gradient of the potential density $\tilde{\rho}^{\Theta}$ is (from McDougall, 1987a)

$$\nabla_n\ln\tilde{\rho}^{\Theta} \approx T_b(P - \tilde{P})\nabla_n\Theta, \tag{37}$$

while from Appendix A of McDougall et al. (2017) we have the following expression for the gradient of $\tilde{\rho}^{\Theta}$ in a $\tilde{\delta}$ surface,

$$\nabla_{\tilde{\delta}}\ln\tilde{\rho}^{\Theta} \approx T_b\nabla_{\tilde{\delta}}[(P - \tilde{P})(\Theta - \widetilde{\Theta})]. \tag{38}$$

The important message that these three equations (36), (37) and (38) deliver is that both $\tilde{\rho}^{\Theta}$ and $\tilde{\delta}$ vary quadratically in space as one moves along a neutral trajectory away from the original location where the ocean properties are $(\tilde{S}_A, \widetilde{\Theta}, \tilde{P})$. In particular, Eq. (38) shows that the originally referenced potential density $\tilde{\rho}^{\Theta}$ varies quadratically with distance along the $\tilde{\delta}$ specific volume anomaly surface, this being the surface along which an adiabatically insulated fluid parcels move. Hence as the limit of a small horizontal displacement is taken, the $\tilde{\delta}$ surface and the $\tilde{\rho}^{\Theta}$ surface coincide (osculate).

There have been three suggested theoretical alternatives to the neutral direction as the direction in which the strong lateral mixing of mesoscale eddies might be directed. The first (chronologically) is the orthobaric surface concept of de Szoeke et al. (2000), the second is the **P**-vector (minimum energy direction) idea of Nycander (2011), and the third is the suggestion of Tailleux (2016a, b) that a neutral surface should be a function of only Absolute Salinity and Conservative Temperature, $\gamma(S_A, \Theta)$. We discuss these three suggestions in the following paragraphs in the chronological order of their publication

de Szoeke et al. (2000) introduced a variable called orthobaric density $\rho_v(\rho, P)$ which is a function of only in-situ density and pressure. McDougall and Jackett (2005b) showed that while it is possible to derive a function of this form to be relatively neutral at all depths in a single ocean basin, it is not possible to do so in both hemispheres because different $\rho_v(\rho, P)$ branches exist on either side of the maximum pressure on a neutral surface. One way of seeing this problem with $\rho_v(\rho, P)$ is from Eq. (37) above. As one proceeds along a neutral path from the outcrop in the South Atlantic to the outcrop in the North Atlantic, the Conservative Temperature increases as does the potential density (referenced to the sea surface) and on some surfaces exceeds $0.1\ \mathrm{kg\ m^{-3}}$. At the two outcrops of a neutral surface in the different hemispheres the pressures are the same, but the in-situ densities are different by $\sim 0.1\ \mathrm{kg\ m^{-3}}$, hence a neutral surface cannot be described by a single-valued function $\rho_v(\rho, P)$. The two different branches of $\rho_v(\rho, P)$ functions bifurcate near the equator where the epineutral gradient of pressure is zero (see McDougall and Jackett, 2005b).

Nycander (2011) studied mixing in the ocean along inclined planes and sought a direction in which a pair of parcels could exchange positions "without encountering a force" (neither a vertical nor a horizontal force), so not require any energy to exchange the parcels. Nycander's **P** direction could, in principle, be different to the neutral tangent plane, however Nycander could not be sure this was the case because the direction of the **P**-vector depends on an unknown reference pressure in the expression for dynamic enthalpy. If this reference pressure is taken to be the in-situ pressure of the parcel exchange, then the **P** surface coincides with the neutral tangent plane. We ask, when considering an individual mixing event, why would any





pressure other than the in-situ pressure be relevant? Because Nycander's approach did not lead to a conclusion as to the direction of lateral mixing in the ocean we interpret this as further support for the long-standing practice (since the 1980s) of defining the neutral direction in terms of the lack of vertical Archimedean buoyant forces, rather than in terms of the changes in gravitational potential energy, or indeed, of any other type of energy. The **P**-vector idea pursued by Nycander (2011) lacks

a physical theorem motivating energetic minimisation as a desirable property to pursue, and in any case, has not led to a conclusion because there is no guidance as to what the reference pressure of the potential energy might be. It is true that the definition of the neutral tangent plane in terms of the lack of Archimedean buoyant forces also lacks a motivating fundamental theorem; instead, it has observational support from the measurements of weak diapycnal mixing in the ocean interior.

      Tailleux (2016a, b) also discuss the mixing by mesoscale eddies and conclude that the surface on which this strong mixing

occurs must be a quasi-material surface, that is, a surface defined by a constant value of a function $\gamma(S_A, \Theta)$ of only Absolute Salinity and Conservative Temperature. Such a function would be formed by averaging globally, or perhaps over just a single ocean basin, but the function that is found by such averaging depends on the region that is included in the average. This is not consistent with the physics of baroclinic instability which depends on the ocean properties in the region of the instability, and certainly not on the details of the water mass distribution in a far-off ocean basin in a different hemisphere. The reason why a

function of the form $\gamma(S_A, \Theta)$ cannot be neutral can also be seen from Eq. (37) above. As one proceeds along a neutral path from the outcrop in the South Atlantic to the outcrop in the North Atlantic, the Conservative Temperature increases as does the potential density (referenced to the sea surface) and on some surfaces exceeds $0.1 \text{ kg m}^{-3}$. This is illustrated in Figures 5 and 9 of McDougall and Jackett (2007). This is the same reason why the "patched isopycnals" of Reid (1994) had the potential density of the northern and southern hemisphere outcrops of the same "patched isopycnal" being different by $\sim 0.1 \text{ kg m}^{-3}$.

The Conservative Temperature and Absolute Salinity on these all-Atlantic neutral surfaces reach maxima at the approximate latitude of the Mediterranean Sea, and at such an extremum of Conservative Temperature on a neutral surface, a new branch of the $\gamma(S_A, \Theta)$ function arises. Hence the properties on such a neutral surface cannot be described by a singled valued function of the $\gamma(S_A, \Theta)$ type. A more detailed explanation of the reasons for the non-neutrality of the $\gamma(S_A, \Theta)$ functional form can be found in McDougall et al. (2017).

**11 Thermobaricity and Cabbeling**

Neglecting the non-conservative source terms (including the term in the dissipation of turbulent kinetic energy, $\varepsilon / \hat{h}_\Theta$), the evolution equation (31) for Conservative Temperature becomes (with $D$ and $K$ being the small-scale isotropic and the epineutral scalar diffusivities respectively)

$$\left. \frac{\partial \Theta}{\partial t} \right|_n + \mathbf{v} \cdot \nabla_n \Theta + e \frac{\partial \Theta}{\partial z} = \gamma_z \nabla_n \cdot (\gamma_z^{-1} K \nabla_n \Theta) + (D\Theta_z)_z, \tag{39}$$

where $\mathbf{v}$ is the thickness-weighted horizontal velocity of density-coordinate averaging, or equivalently the horizontal Temporal-Residual-Mean (TRM) velocity, $e$ is the mean vertical velocity through the locally referenced potential density





surface, and $\Theta$ is the thickness-weighted Conservative Temperature of density coordinate averaging. The terms in the vertical gradient of locally referenced potential density, $\gamma_z$, and its reciprocal, account for the lateral gradient of the thickness between density surfaces along which the lateral diffusivity $K$ acts. The lateral gradient operator $\nabla_n$ measures the lateral variations of

properties in the locally referenced potential density surface which osculates with the neutral tangent plane. The averaging procedure to obtain this TRM-mean evolution equation was derived by McDougall and McIntosh (2001) and is also explained in section A21 of the TEOS-10 Manual (IOC et al., 2010). The corresponding evolution equation for Absolute Salinity is (also ignoring the biogeochemical source term of Absolute Salinity, which for large-scale oceanography, is not insignificant)

$$\left.\frac{\partial S_A}{\partial t}\right|_n + \mathbf{v}\cdot\nabla_n S_A + e\frac{\partial S_A}{\partial z} \;=\; \gamma_z\nabla_n\cdot(\gamma_z^{-1}K\nabla_n S_A) + \left(DS_{A_z}\right)_z. \tag{40}$$

It is instructive to take linear combinations of these evolution equations for $\Theta$ and $S_A$ with the combinations chosen to first eliminate the dianeutral velocity $e$, and then to eliminate the temporal and lateral advection terms (using the temporal and spatial neutral relationships Eq. 34), leading to the following two equations (where the cabbeling coefficient is defined by $C_b \equiv \hat\alpha_\Theta + 2(\hat\alpha/\hat\beta)\hat\alpha_{S_A} - (\hat\alpha/\hat\beta)^2\hat\beta_{S_A}$ where the over-hats indicate that these variables are functions of $(S_A, \Theta, P)$).

$$\left.\frac{\partial\Theta}{\partial t}\right|_n + \mathbf{v}\cdot\nabla_n\Theta \;=\; \gamma_z\nabla_n\cdot(\gamma_z^{-1}K\nabla_n\Theta) + KgN^{-2}\Theta_z(C_b\nabla_n\Theta\cdot\nabla_n\Theta + T_b\nabla_n\Theta\cdot\nabla_n P) + D\beta gN^{-2}\Theta_z^3\frac{d^2 S_A}{d\Theta^2}, \tag{41}$$

and

$$e\,g^{-1}N^2 \;=\; -K(C_b\nabla_n\Theta\cdot\nabla_n\Theta + T_b\nabla_n\Theta\cdot\nabla_n P) + \alpha(D\Theta_z)_z - \beta\left(DS_{A_z}\right)_z. \tag{42}$$

These equations were derived by McDougall (1987a, b) and a discussion of them can also be found in sections A22 and A23 of IOC et al. (2010). Eq. (42) shows that the dianeutral velocity is not a separate physical process but rather is a consequence of the processes of lateral and small-scale isotropic mixing as parameterized with the diffusivities $K$ and $D$.

When the dianeutral velocity is eliminated to obtain Eq. (41) one finds that the small-scale mixing process, $D$, only affects epineutral changes (in time and/or space) in proportion to the curvature $d^2 S_A/d\Theta^2$ of the vertical water column on the $S_A - \Theta$ diagram. In the other case, when the epineutral gradients are eliminated, one finds in Eq. (42) that the dianeutral velocity is caused not only by the small-scale isotropic mixing process but also by $-K(C_b\nabla_n\Theta\cdot\nabla_n\Theta + T_b\nabla_n\Theta\cdot\nabla_n P)$ where the epineutral mixing acts in conjunction with the nonlinear nature of the equation of state, resulting in two contributions, cabbeling

and thermobaricity, to dianeutral advection.

The thermobaricity and cabbeling processes can be understood with the aid of Figure 10. Epineutral mixing involves first the movement of fluid parcels in an adiabatic and isentropic manner, followed by the intimate mixing between them. During the adiabatic and isohaline movements the two seawater parcels move along specific volume anomaly surfaces (see the discussion above). Consider two fluid parcels labelled A and B in Figure 10a that have been chosen because they get to meet

and mix subsequently at physical location D. Parcels A and B were originally on a neutral trajectory along which there are variations of pressure (slope) and Conservative Temperature, and as parcel A is moved to the right, its constant value of



Conservative Temperature is soon less than that on the neutral trajectory. This means that the insulated parcel A has a larger compressibility than the fluid surrounding it, and as it moves further to the right it sinks to be further separated from the neutral trajectory. This can be seen in Eq. (36); the insulated parcel A has a constant (zero) value of $\tilde{\delta}$, but the value of $\tilde{\delta}$ on the neutral

trajectory increases quadratically to the right from the original location A. The opposite thing happens to parcel B, and the two parcels A and B soon meet at a location where the ocean's properties have properties D. Up to this point in the process no intimate mixing has occurred, and in principle, the movements could be reversed. Upon mixing, the density of the mixed fluid is increased so that the potential density referenced to the pressure of parcel D increases and the mixture has properties E (on Figure 10b) and sinks to location E on Figure 10a. This part of the process is called cabbeling, and the intimate mixing has

also made the dianeutral advection thermobaricity irreversible (permanent).

Figure 10 is an explanation of the thermobaricity and cabbeling processes based on the advection and mixing of two fluid parcels. Klocker and McDougall (2010a) have shown how the total dianeutral transport (in Sv) across an oceanic epineutral front can be quantified in terms of the epineutral flux of temperature, $-K\nabla_n\Theta$, and the total epineutral differences of Conservative Temperature and of pressure across the front. In this way the area-integrated effect of thermobaricity and

cabbeling do not depend on having to estimate the epineutral property gradients such as $\nabla_n\Theta$, or the epineutral diffusivity $K$, but rather depend on the epineutral fluxes themselves, and the epineutral temperature and pressure differences.

Several studies such as Iudicone et al. (2008), Klocker and McDougall (2010a) and Groeskamp et al. (2016) have quantified the magnitudes of thermobaricity and cabbeling in the global ocean and have shown that they are the dominant contributor to dianeutral advection in the Southern ocean.

**12 Neutral helicity and the ill-defined nature of neutral surfaces**

Equation (34) above defines the neutral tangent plane in terms of the compensating gradients of Absolute Salinity and Conservative Temperature, $\alpha\nabla_n\Theta - \beta\nabla_n S_A = 0$, or equivalently, $\kappa\nabla_n P - \nabla_n\ln\rho = 0$. The neutral tangent plane can be found at each point in the ocean, but does this guarantee that these neutral tangent planes can be linked together to find a well-defined surface? Let us begin our discussion by assuming that these well-defined neutral surfaces exist. In that case the integral along

a series on neutral tangent planes around any closed path in longitude-latitude space will arrive back at the start at the same height as at the beginning of the loop. This means that the closed integral of Absolute Salinity around a loop in the surface is zero, that is, $\oint dS_A = 0$, and it follows that the closed integral around the same loop along the neutral tangent planes, $\oint(\alpha/\beta)d\Theta$, must also be zero. This is guaranteed to hold if $\alpha/\beta$ is a function of salinity and temperature only, but since $\alpha/\beta$ is also a function of pressure, this closed integral will not in general be zero, and so our assumption that the neutral surface

exists has been proven untrue. Figure 1 of McDougall and Jackett (1988) shows how the pressure dependence of $\alpha/\beta$ causes neutral surfaces to be ill-defined.

Here we explain the ill-defined nature of neutral surfaces by examining the property needed of the normal to the neutral tangent plane for a neutral surface to be well-defined. The (three dimensional) normal to the neutral tangent plane is in the



direction $\alpha\nabla\Theta - \beta\nabla S_A = \kappa\nabla P - \nabla\ln\rho$ whose vertical component has magnitude $g^{-1}N^2$. The neutral tangent plane and its

normal, $\alpha\nabla\Theta - \beta\nabla S_A$, exist at every point in space and one might think that this is sufficient to ensure that all the little tangent planes would join up to become a well-defined surface. But this is not the case. Rather, it can be shown (McDougall and Jackett, 1988 and McDougall and Jackett 2007) that the scalar product of the normal $\alpha\nabla\Theta - \beta\nabla S_A$ with its curl must be zero everywhere in order for all the neutral tangent planes to join up and describe a surface. This triple scalar product is called the neutral helicity, $H$,

$$
\begin{aligned}
H &\equiv (\alpha\nabla\Theta - \beta\nabla S_A) \cdot \nabla \times (\alpha\nabla\Theta - \beta\nabla S_A) \\
&= \beta T_b \nabla P \cdot \nabla S_A \times \nabla\Theta \\
&= g^{-1}N^2 T_b \nabla_n P \times \nabla_n\Theta \cdot \mathbf{k} \\
&= P_z \beta T_b \nabla_p S_A \times \nabla_p\Theta \cdot \mathbf{k}.
\end{aligned}
\tag{43}
$$

The last three parts of this equation have been found by expanding the definition of $H$ in the first line, and $\mathbf{k}$ is the unit vertical

vector. Each of these expressions contain the thermobaric parameter $T_b \equiv \alpha_P - (\alpha/\beta)\beta_P = \beta(\alpha/\beta)_P$ which again lays the blame for a non-zero neutral helicity on the fact that $\alpha/\beta$ is a function of pressure. We can discuss this equation in relation to Figure 11. First, we know from the definition of the neutral tangent plane definition that $\alpha\nabla_n\Theta - \beta\nabla_n S_A = 0$ so the neutral tangent plane must contain the line of constant temperature and salinity, $\nabla\Theta \times \nabla S_A$. Similarly, we know that $\kappa\nabla_n P - \nabla_n\ln\rho = 0$ so the neutral tangent plane must also contain the line $\nabla P \times \nabla\rho$. Figure 11 shows the lines $\nabla\Theta \times \nabla S_A$ and $\nabla P \times \nabla\rho$, the

neutral tangent plane (in red) and five other coloured planes.

The second line of Eq. (43), $H = \beta T_b \nabla P \cdot \nabla S_A \times \nabla\Theta$, can be understood from Figure 11 as requiring that if the neutral helicity $H$ is to be zero then the two lines $\nabla\Theta \times \nabla S_A$ and $\nabla P \times \nabla\rho$ must coincide. That is, if $H$ is to be zero $\nabla\Theta \times \nabla S_A$ must lie in the plane of constant pressure; approximately we can say that $\nabla\Theta \times \nabla S_A$ must be a horizontal vector with no vertical component. The third line of Eq. (43) says that the two-dimensional gradients of pressure and temperature in the neutral

tangent plane must be parallel to each other if neutral helicity is to be zero. The fourth line of Eq. (43) says that the two-dimensional gradients of salinity and temperature in the pressure surface must be parallel to each other if neutral helicity is to be zero. Each of these constraints that result from a requirement that neutral helicity be zero seem quite restrictive, but it turns out that most of the ocean has small values of neutral helicity, with the epineutral gradients of pressure and temperature, $\nabla_n P$ and $\nabla_n\Theta$ being close to parallel. Each of these geometrical interpretations of neutral helicity have been explored and illustrated

with oceanographic data in McDougall and Jackett (1988) and (2007).

Note that neutral helicity can be expressed in terms of the thermal wind (the vertical gradient of the Eulerian mean horizontal velocity) by

$$
H = \rho T_b f \mathbf{v}_z \cdot \nabla_n\Theta ,
\tag{44}
$$

showing that neutral helicity is proportional to the component of the thermal wind, $\mathbf{v}_z$, in the direction of the epineutral

temperature gradient, $\nabla_n\Theta$.





The same geometrical combination of two-dimensional temperature, salinity and pressure gradients, $\nabla_n P \times \nabla_n \Theta \cdot \mathbf{k}$, that appears in the expression for neutral helicity also occurs in the expression for the mean absolute velocity in terms of the ocean's hydrography, but without being multiplied the thermobaric coefficient. The geostrophic balance can be exploited to find the following expression for the mean Eulerian velocity $\bar{\mathbf{v}}$ (as opposed to the temporal residual-mean velocity $\mathbf{v}$ in Eq. (41))


$$\bar{\mathbf{v}} = \left\{ \frac{N^2}{fg\rho} \frac{\mathbf{k} \cdot \nabla_n P \times \boldsymbol{\tau}}{\phi_z} - \frac{v_z^\perp}{\phi_z} \right\} \boldsymbol{\tau} \times \mathbf{k} + v^\perp \boldsymbol{\tau} ,\qquad(45)$$

(McDougall, 1995 and IOC et al., 2010) where $\boldsymbol{\tau}$ is the unit epineutral temperature gradient vector, $\nabla_n\Theta / |\nabla_n\Theta|$, $\phi_z$ is the rate at which the direction of $\nabla_n\Theta$ changes in the vertical (in radians per meter) and $v^\perp$ is the component of $\bar{\mathbf{v}}$ in the $\boldsymbol{\tau}$ direction; this component being caused by mixing processes (as in Eq. (41) above). Estimates of the absolute velocity usually come from performing an inverse study, and in fact Eq. (45) is the result of such a method since it uses the geostrophic balance

equation at two heights separated by the infinitesimal distance d$z$ in the vertical. Equation (45) implies that neutral helicity needs to be non-zero in order for the ocean to have mean motion (apart the mean velocity caused by mixing processes). The ocean seems able to satisfy this requirement while having what seems like small neutral helicity in most places (see the following sub-section 12.1).

**12.1 The ocean is quite empty in salinity-temperature-pressure space**

Here we explore the constraints on the ocean hydrography if neutral helicity $\nabla P \cdot \nabla S_A \times \nabla \Theta$ were zero everywhere in the ocean. It can be shown that if $P(x, y, z)$, $S_A(x, y, z)$ and $\Theta(x, y, z)$ everywhere obey $\nabla P \cdot \nabla S_A \times \nabla \Theta = 0$, then all the pressure, salinity and temperature data lie on a single surface in $S_A - \Theta - P$ space (McDougall and Jackett, 2007). This theoretical result prompted the visualization of the salinity-temperature-pressure data of the global ocean by rotating this ocean data in $S_A - \Theta - P$ space. This revealed that the global ocean is rather "hollow" in this space, as is illustrated in Figure 12.

Figure 12(a) shows a particular view in $S_A - \Theta - P$ space of all the hydrography of the South and North Atlantic oceans. The blue data from the South Atlantic is clearly separated from the North Atlantic data in red. It is much more obvious to see that the data lies close to a single surface when viewing the data as it rotates in $S_A - \Theta - P$ space on a computer screen, thus viewing the data from many different angles. Another way of making this point is to display the hydrographic data on the two-dimensional $S_A - \Theta$ plot at just one pressure, and an example is shown in Figure 12(b). This two-dimensional cut through

the three-dimensional $S_A - \Theta - P$ data clearly demonstrates the emptiness of the ocean's hydrography in $S_A - \Theta - P$ space. At the pressure of 1010 dbar (Fig. 12(b)) the data snakes around on the $S_A - \Theta$ diagram with the warmest and saltiest data being the signature of Mediterranean Water in the North Atlantic. If the data fell exactly on a single curved line then the neutral helicity would be exactly zero everywhere at this pressure and neutral surfaces would be well-defined in this vicinity. This is not totally the case, but the data fills only a few percent of the area described by connecting every data point to every

other point on Fig. 12(b). It is not immediately obvious whether filling out only a few percent of the area on such a diagram represents a very small area or not, but the next sub-section attempts to address this question by concentrating on the



consequences of (1) having lateral mixing occurring along surfaces that differ from the neutral tangent plane, and (2) having the mean vertical advection that occurs because the lateral flow is forced to move along an approximately neutral surface rather than along the neutral tangent plane.

## 12.2 Approximately neutral surfaces

There are many different types of surfaces that have been designed with the aim of being approximately neutral. Examples are potential density surfaces, patched potential density surfaces (Reid, 1994), Neutral Density surfaces, and some of the purpose-built algorithms specifically designed to approximate neutrality. The Neutral Density algorithm of Jackett and McDougall (1985) operates by neutrally relating a given hydrographic observation to a pre-labelled global atlas. Having labelled a data set with Neutral Density, an iso-surface of Neutral Density can be formed. The Neutral Density algorithm has not yet been updated from EOS-80 to TEOS-10, nor does it yet operate north of 60°N in the North Atlantic. Fixing these deficiencies requires a new labelled reference data set of neutral density values in a global atlas that extends into the Arctic Ocean.

Another type of approximately neutral surface is called an "$\varpi$ surface" (Klocker et al., 2009) which is formed iteratively from an initial surface which could be a potential density surface or even an isobaric surface. In this initial surface the gradients of Absolute Salinity and Conservative Temperature do not completely compensate each other in terms of their effects of density. That is, $\alpha\nabla_a\Theta - \beta\nabla_n S_A \neq 0$, in comparison with the neutral tangent plane in which $\alpha\nabla_n\Theta - \beta\nabla_n S_A$ is zero. The method works using a two-step iterative procedure. In the first step the scalar function of longitude and latitude $\Phi(x, y)$ is found which minimizes the integral of

$$(\alpha\nabla_a\Theta - \beta\nabla_a S_A + \nabla_a\Phi) \cdot (\alpha\nabla_a\Theta - \beta\nabla_a S_A + \nabla_a\Phi), \tag{46}$$

over the area of the initial surface. $\Phi(x, y)$ is then interpreted as an increment of the natural logarithm of locally referenced potential density and the second step finds the increment of height on the $(x, y)$ vertical cast corresponding to this increment $\Phi(x, y)$ in $\ln\rho^l$. The updated surface is then taken to be the original height of the surface plus this height increment. The process is then repeated with this updated surface as the initial condition. Stanley et al. (2021) have made this process computationally efficient using a Poisson equation solver, converging to the final $\varpi$ surface after just two or three iterations after starting with a potential density surface, and the method includes the "wetting" at new $(x, y)$ locations as each successive iteration adjusts the height of the surface and hence its ability to talk neutrally to adjacent locations.

Lang et al. (2023) have presented additional variations of the $\varpi$ surface methodology by minimizing not only the gradient of locally referenced potential density in the neutral tangent plane but also the component of the three-dimensional velocity vector through the final surface due to it not exactly coinciding with the neutral tangent plane. This method needs knowledge of the horizontal velocity at each location on the surface and so it is suitable for use with ocean model output.

Klocker et al. (2010b) quantified the vertical advection through their $\varpi$ surfaces due to the ill-defined nature of neutral surfaces and found that while the area-average of this dia-surface velocity is small, locally it can be as large as the canonical





$10^{-7}$ m s$^{-1}$. By penalising not only the slope error between the surface and the neutral tangent plane but also the spurious dia-
surface flow, Lang et al. (2023) were able to reduce the root-mean-square dia-surface velocity to a few by $10^{-9}$ m s$^{-1}$ which
makes such surfaces suitable for the conduct of oceanographic inverse studies.

We have now discussed two processes that involve the thermobaric coefficient that lead to mean dianeutral motion, namely
(i) thermobaricity and (ii) the mean dianeutral motion caused by the path dependent nature of neutral surfaces. Note
thermobaricity depends on the parallel (scalar product) of the epineutral $\Theta$ and $P$ gradients, namely $\nabla_n P \cdot \nabla_n \Theta$, whereas the
path dependent nature of neutral surfaces depends on the cross product $\nabla_n P \times \nabla_n \Theta \cdot \mathbf{k}$.

## 13 Neutral Surface Potential Vorticity

Maps of potential vorticity on neutral surfaces are used to deduce the direction of the horizontal circulation of the ocean,
and for this purpose, $fN^2$, the product of the Coriolis frequency and the square of the buoyancy frequency, is often used as an
approximation to planetary potential vorticity. In this section we quantify the error made in this approximation.

For the present purposes we will take the ocean hydrography to be in steady-state, Neutral Helicity will be assumed
sufficiently small that the existence of neutral surfaces is a good approximation, and we seek the integrating factor $b = b(x, y, z)$ which allows the construction of Neutral Density surfaces ($\gamma$ surfaces) according to (with $\alpha$ being the thermal
expansion coefficient of seawater, $\beta$ is the saline contraction coefficient and $\kappa$ is the adiabatic compressibility)

$$\nabla \ln \gamma = b(\beta \nabla S_A - \alpha \nabla \Theta) = b(\nabla \ln \rho - \kappa \nabla P). \tag{47}$$

We will ignore the relative vorticity in comparison to the Coriolis frequency, $f$, so that this section discusses planetary
relative vorticity, but the equations we develop apply equally when relative vorticity is included. Dianeutral advection is
caused by small-scale turbulent mixing processes, including double-diffusive convection, as well as by thermobaricity and
cabbeling. These contributions to dianeutral motion and to the production of potential vorticity are not discussed here. Rather,
we note that the strong lateral mixing and advection in the ocean occurs along neutral surfaces so that Neutral Surface Potential
Vorticity (NSPV) is defined as being proportional to the Coriolis frequency $f$ divided by the vertical distance between adjacent
neural surfaces. Specifically, following McDougall (1988) and Straub (1999), NSPV is defined by

$$\text{NSPV} = -fg(\ln \gamma)_z . \tag{48}$$

If the equation of state were linear this would be equal to $fN^2$, and this approximation is often made in the literature. Here
we concentrate on the influence of the nonlinear nature of the equation of state of seawater on the calculation of NSPV by
finding expressions for the integrating factor

$$b = \frac{\text{NSPV}}{fN^2} = \frac{-g(\ln \gamma)_z}{N^2}. \tag{49}$$

We begin by considering the simplest case where neutral surfaces are horizontal (zero epineutral gradient of pressure) and
where there are also no spatial variations of Absolute Salinity and Conservative Temperature on each neutral surface. In this





situation the square of the buoyancy frequency, $N^2$, is constant along the neutral surfaces and the Neutral Surface Planetary

Potential Vorticity (NSPV) is simply proportional to the Coriolis frequency $f$ times $N^2$, and the integrating factor $b$ is a

constant. We will then consider two other special cases before deriving a general expression for the epineutral gradient of the

integrating factor, $b$, which relates $fN^2$ to NSPV. The first of these two special cases is when the neutral surfaces are

horizontal but there are epineutral gradients of temperature (and salinity), while in the second special case there are no

epineutral gradients of salinity or temperature, but the neutral surfaces are not horizontal.

**13.1 The special case $\nabla_n P = 0$**

Here we consider the variable $\tilde{r}$ defined by

$$\tilde{r} \equiv \frac{\rho(S_A, \Theta, P)}{\rho(\tilde{S}_A, \widetilde{\Theta}, P)} = \frac{v(\tilde{S}_A, \widetilde{\Theta}, P)}{v(S_A, \Theta, P)}. \tag{50}$$

where the over-tilde on $\tilde{S}_A$ and $\widetilde{\Theta}$ indicates the values of these variables at a chosen reference location on a specific neutral

surface. We call $\tilde{r}$ the "specific volume ratio" since it has many characteristics in common with the well-known "specific

volume anomaly" defined by $\tilde{\delta} \equiv v(S_A, \Theta, P) - v(\tilde{S}_A, \widetilde{\Theta}, P)$. Taking the epineutral gradient of $\tilde{r}$ we find (since $\alpha\nabla_n\Theta = \beta\nabla_n S_A$ on each neutral surface)

$$\nabla_n \ln \tilde{r} = g\rho[\kappa(S_A, \Theta, P) - \kappa(\tilde{S}_A, \widetilde{\Theta}, P)]\nabla_n P, \tag{51}$$

(where $\kappa$ is the adiabatic compressibility of seawater). The criterion $\nabla_n P = 0$ is now assumed to hold in some finite volume

in space, that is, on neutral surfaces both above and below the central surface we are studying, so that in our special case of

$\nabla_n P = 0$ we find that $\tilde{r}$ does not vary along neutral surfaces. In this case the integrating factor, $b$, which relates the vertical

gradient of a neutral density variable to $N^2$ is

$$b = \frac{\text{NSPV}}{fN^2} = \frac{-g(\ln \gamma)_z}{N^2} = \tilde{b}\left[\frac{-g(\ln \tilde{r})_z}{N^2}\right], \qquad\qquad \text{ocean with } \nabla_n P = 0 \tag{52}$$

since in this region of space successive neutral surfaces coincide with $\tilde{r}$ surfaces so that $(\ln \tilde{r})_z$ is proportional to $(\ln \gamma)_z$. Here

$\tilde{b}$ is the value of the integrating factor at the reference location $(\tilde{S}_A, \widetilde{\Theta}, \tilde{P})$ where $-g(\ln \tilde{r})_z = N^2$. We now seek an equation

for the epineutral gradient of the integrating factor $b$ and we begin by first taking the vertical gradient of $\ln \tilde{r}$, obtaining

$$-(\ln \tilde{r})_z = g^{-1}N^2 + g\rho[\kappa(S_A, \Theta, P) - \kappa(\tilde{S}_A, \widetilde{\Theta}, P)], \tag{53}$$

and then taking the epineutral gradient of this equation, to find that in this $\nabla_n P = 0$ case where $(\ln \tilde{r})_z$ is constant along neutral

surfaces (recall that since $\nabla_n P = 0$ in this region of space, the thickness between successive neutral surfaces is essentially

constant), $b = \tilde{b}[-gN^{-2}(\ln \tilde{r})_z]$ obeys

$$\nabla_n \ln b = -\rho g^2 N^{-2}T_b\nabla_n\Theta. \qquad\qquad \text{ocean with } \nabla_n P = 0 \tag{54}$$



This derivation has used the fact that in this special case $\nabla_n N^2 = \rho g^2 T_b \nabla_n \Theta$ which can be shown to hold by expanding its left-hand side in the form $g\nabla_n(\alpha\Theta_z - \beta S_{A_z})$. Since the specific volume ratio $\tilde{r}$ is constant along each neutral surface in this case, the neutral surface potential vorticity (NSPV) is proportional to the Coriolis frequency times the vertical gradient of $\ln \tilde{r}$, and the epineutral variations of the integrating factor in this case reflects that $f N^2$ is not proportional to NSPV.

Spatially integrating Eq. (54) we find the following expression for the integrating factor in this special situation,

$$b = \frac{\text{NSPV}}{f N^2} = \frac{-g(\ln\gamma)_z}{N^2} = \tilde{b}\left[\frac{-g(\ln\tilde{r})_z}{N^2}\right] = \tilde{b}\exp\{-\int \rho g^2 N^{-2} T_b \nabla_n\Theta \cdot d\boldsymbol{l}\}. \qquad \text{ocean with } \nabla_n P = \boldsymbol{0} \qquad (55)$$

This result seems surprising because it relates $N^2$ (since $g(\ln\tilde{r})_z$ is constant along the neutral surface) at any location on the neutral surface to a lateral integral of $\rho g^2 N^{-2} T_b \nabla_n\Theta$ along the surface. We discuss this aspect of Eq. (113) in relation to Figure 13 below.

## 13.2 The special case $\nabla_n\Theta = 0$

Consider now a region of an ocean in which the epineutral gradients of salinity and temperature are zero, that is, $\nabla_n\Theta = \boldsymbol{0}$. This criterion is assumed to hold in some finite volume in space, that is, on neutral surfaces both above and below the central surface we are studying. In this region the ocean $S_A - \Theta$ properties lie on a single (possibly curved) line in $S_A - \Theta$ space, and the neutral helicity is zero. In this case potential density $\tilde{\rho}^\Theta \equiv \rho(S_A, \Theta, \tilde{P})$ referenced to any fixed pressure, $\tilde{P}$, is constant

along each neutral surface since

$$\nabla_n \ln \tilde{\rho}^\Theta = \beta(S_A, \Theta, \tilde{P})\nabla_n S_A - \alpha(S_A, \Theta, \tilde{P})\nabla_n\Theta = \beta(S_A, \Theta, \tilde{P})\left[\frac{\alpha}{\beta}(S_A, \Theta, P) - \frac{\alpha}{\beta}(S_A, \Theta, \tilde{P})\right]\nabla_n\Theta. \qquad (56)$$

Since potential density is constant along each neutral surface in this case, the neutral surface potential vorticity (NSPV) is proportional to the Coriolis frequency times the vertical gradient of $\ln \tilde{\rho}^\Theta$. However, this does not mean that the integrating factor $b$ is unity. Rather, the integrating factor in this special $\nabla_n\Theta = \boldsymbol{0}$ case is given by

$$b = \frac{\text{NSPV}}{f N^2} = \frac{-g(\ln\gamma)_z}{N^2} = \tilde{b}\left[\frac{-g(\ln\tilde{\rho}^\Theta)_z}{N^2}\right], \qquad \text{ocean with } \nabla_n\Theta = \boldsymbol{0} \qquad (57)$$

and $N^2$ does vary along the neutral surfaces. We now seek an equation for the epineutral gradient of the integrating factor $b$ and we begin by first taking the vertical gradient of $\tilde{\rho}^\Theta$, obtaining (with $\tilde{\alpha}$ being shorthand for $\alpha(S_A, \Theta, \tilde{P})$ and $\tilde{\beta}$ standing for $\beta(S_A, \Theta, \tilde{P})$)

$$-(\ln\tilde{\rho}^\Theta)_z = g^{-1}N^2 - (\alpha - \tilde{\alpha})\Theta_z + (\beta - \tilde{\beta})S_{A_z}, \qquad (58)$$

followed by taking the epineutral gradient of this equation, to find

$$\nabla_n \ln b = -g N^{-2}(\alpha_P \Theta_z - \beta_P S_{A_z})\nabla_n P, \qquad \text{ocean with } \nabla_n\Theta = \boldsymbol{0} \qquad (59)$$



where $b = \tilde{b}[-gN^{-2}(\ln \tilde{\rho}^\Theta)_z]$ in this $\nabla_n\Theta = 0$ case. This derivation relies first on vertically stretching each water column so that the thickness between adjacent neutral surfaces is spatially uniform (since this stretching does not affect the ratio $-gN^{-2}(\ln \tilde{\rho}^\Theta)_z$) and then deriving the relationship $\nabla_n N^2 = g(\alpha_P\Theta_z - \beta_P S_{A_z})\nabla_n P$ by expanding its left-hand side in the form $g\nabla_n(\alpha\Theta_z - \beta S_{A_z})$. Since the neutral surface potential vorticity (NSPV) in this case is proportional to the Coriolis frequency times $(\ln \tilde{\rho}^\Theta)_z$, the epineutral variations of the integrating factor in this case again reflects that $fN^2$ is not proportional to NSPV.

Spatially integrating Eq. (59) we find the following expression for the integrating factor in this special situation,

$$b = \frac{\text{NSPV}}{fN^2} = \frac{-g(\ln \gamma)_z}{N^2} = \tilde{b}\left[\frac{-g(\ln \tilde{\rho}^\Theta)_z}{N^2}\right] = \tilde{b}\exp\left\{-\int gN^{-2}(\alpha_P\Theta_z - \beta_P S_{A_z})\nabla_n P \cdot d\boldsymbol{l}\right\}. \quad \text{ocean with } \nabla_n\Theta = 0 \quad (60)$$

This result seems surprising because it relates a vertical integral of $N^2$ between a pair of closely spaced neutral surfaces at any location on the neutral surface to a lateral integral of $gN^{-2}(\alpha_P\Theta_z - \beta_P S_{A_z})\nabla_n P$ along the surface from the reference location, in this special $\nabla_n\Theta = 0$ case. We discuss this aspect of Eq. (60) in relation to Figure 13 below.

One example of this $\nabla_n\Theta = 0$ special case is when the ocean resembles a lake in that there are no variations of salinity in any direction in space. Then the neutral surfaces are surfaces of constant Conservative Temperature and the fact that the integrating factor is not constant along a neutral surface again reflects how $N^2$ is adversely affected by the thermobaric term in the equation of state.

### 13.3 The general expression for the integrating factor

Taking the curl of Eqn. (47) gives

$$\nabla\ln b \times (\kappa\nabla P - \nabla\ln\rho) = -\nabla\kappa \times \nabla P. \quad (61)$$

The bracket on the left-hand side is normal to the neutral tangent plane, pointing in the direction $\mathbf{n} = -\nabla_n z + \mathbf{k}$ and is $g^{-1}N^2(-\nabla_n z + \mathbf{k})$. Taking the component of Eq. (61) in the direction of the normal to the neutral tangent plane, $\mathbf{n}$, we find (using the equalities $\kappa_{S_A} = \beta_P$ and $\kappa_\Theta = -\alpha_P$)

$$0 = \nabla\kappa \times \nabla P \cdot \mathbf{n} = (\nabla_n\kappa + \kappa_z\mathbf{n}) \times (\nabla_n P + P_z\mathbf{n}) \cdot \mathbf{n}$$

$$= \nabla_n\kappa \times \nabla_n P \cdot \mathbf{k} = (\kappa_{S_A}\nabla_n S_A + \kappa_\Theta\nabla_n\Theta) \times \nabla_n P \cdot \mathbf{k} \quad (62)$$

$$= T_b\nabla_n P \times \nabla_n\Theta \cdot \mathbf{k} = gN^{-2}H^n,$$

which simply says that the neutral helicity $H^n \equiv g^{-1}N^2 T_b\nabla_n P \times \nabla_n\Theta \cdot \mathbf{k}$, must be zero for the dianeutral component of Eq. (61) to hold.

Writing $\nabla b$ as $\nabla_n b + b_z\mathbf{n}$, Eq. (120) becomes

$$g^{-1}N^2\nabla_n\ln b \times (-\nabla_n z + \mathbf{k}) = -P_z\nabla_p\kappa \times (-\nabla_p z + \mathbf{k}) \quad (63)$$



where $\nabla P = P_z(-\nabla_p z + \mathbf{k})$ has been used, $(-\nabla_p z + \mathbf{k})$ being normal to the isobaric surface. Concentrating on the

horizontal components of this equation we see that $g^{-1}N^2\nabla_n \ln b = -P_z\nabla_p \kappa$, and using the hydrostatic equation $P_z = -g\rho$

gives the tantalizingly simple relationship

$$\nabla_n \ln b = \rho g^2 N^{-2}\nabla_p \kappa. \tag{64}$$

We now write $\nabla_p \kappa$ in Eq. (64) as $\nabla_n \kappa + \rho^{-1}g^{-1}\kappa_z\nabla_n P$ (which has used the hydrostatic equation $P_z = -g\rho$ and the

relationship $[\nabla_p z - \nabla_n z] = \rho^{-1}g^{-1}\nabla_n P$) and then expand both the epineutral and vertical gradients of $\kappa$ in terms of its

thermodynamic partial derivatives $\kappa_{S_A}, \kappa_\Theta, \kappa_P$ and the corresponding spatial gradients of salinity, temperature and pressure.

This leads to [noting that during the expansion the two terms in $\kappa_P\nabla_n P$ cancel, and that the definition of the thermobaric

parameter is $T_b = \alpha_P - (\alpha/\beta)\beta_P$]

$$\nabla_n \ln b = -\rho g^2 N^{-2}T_b\nabla_n\Theta - gN^{-2}(\alpha_P\Theta_z - \beta_P S_{A_z})\nabla_n P. \tag{65}$$

This form of the expression for $\nabla_n \ln b$ has expressed the isobaric gradient expression $\rho g^2 N^{-2}\nabla_p \kappa$ of Eq. (64) into

contributions from the epineutral variations of spice ($\nabla_n\Theta$) and those due to the slope of the neutral surface ($\nabla_n P$). This

equation can be spatially integrated from a location on a given approximately neutral surface where the Absolute Salinity,

Conservative Temperature and absolute pressure are $(\tilde{S}_A, \tilde{\Theta}, \tilde{P})$ obtaining

$$b = \frac{\text{NSPV}}{fN^2} = \frac{-g(\ln\gamma)_z}{N^2} = \tilde{b}\exp\{-\int \rho g^2 N^{-2}T_b\nabla_n\Theta\cdot d\boldsymbol{l}\}\exp\{-\int gN^{-2}(\alpha_P\Theta_z - \beta_P S_{A_z})\nabla_n P\cdot d\boldsymbol{l}\}, \tag{66}$$

where $\tilde{b}$ is the value of the integrating factor at $(\tilde{S}_A, \tilde{\Theta}, \tilde{P})$ on this approximately neutral surface, and the integrals are performed

along this surface from the $(\tilde{S}_A, \tilde{\Theta}, \tilde{P})$ location. Since we have assumed zero neutral helicity, $\nabla_n\Theta$ and $\nabla_n P$ have been assumed

to be parallel. This expression (66) was originally found by McDougall (1988) and was derived in the above compact manner

in the TEOS-10 manual (IOC et al., 2010).

An example of Eq. (66) in action is illustrated in Figure 13. Panel (a) shows a vertical cross section of five neutral surfaces

between vertical casts A and B. Surfaces 2 and 4 are depicted at an initial time by dashed lines and by full lines at a later time,

while surfaces 1, 3 and 5 are in full lines in both epochs. Panels (b) and (c) show the cross section through these five neutral

surfaces for both epochs on the $S_A - \Theta$ and $\Theta - P$ diagrams respectively. Between the two epochs surfaces 2 and 4 have

undergone some vertical heaving (exaggerated in the figure) so that the vertical distance between these two surfaces is

increased in the later epoch compared with the first epoch. The integrand of the second exponential integral expression in Eq.

(66) will be approximately the same in the two epochs because along surface 3 both $g^{-1}N^2$ and $(\alpha_P\Theta_z - \beta_P S_{A_z})$ will be

affected by the heave in approximately the same proportion so their ratio will not be affected. The integrand in the first

exponential expression is directly affected by the larger vertical distance between surfaces 2 and 4 in the later epoch compared

with the first, with $N^{-2}$ being larger at the later time, resulting in a larger negative exponent of the exponential, so causing the





integrating factor $b$ at cast B to be less in the later epoch compared with the first. This is consistent with the vertical spacing
between points 2' and 4' being greater on cast B than the vertical distance between points 2 and 4. The different vertical
locations of points 2 and 2' (and also between 4 and 4') can also be understood as a consequence of neutral helicity. Integrating
Eq. (37) along neutral trajectory 2 from point 2 on cast B to cast to pint 2 on cast A and then back to cast B along the dashed
neutral trajectory to point 2' on cast B, one finds that the difference in potential density can be deduced from the shaded area
in Fig. 12(c) according to

$$\Delta(\ln \tilde{\rho}^{\Theta}) \approx \oint T_b(P - \tilde{P}) \, d\Theta. \tag{67}$$

### 13.4 Discussion of the expressions for the integrating factor

In the above we have found in the special case when $\nabla_n P = \mathbf{0}$, that $\nabla_n \ln b = -\rho g^2 N^{-2} T_b \nabla_n \Theta$ and when $\nabla_n \Theta = \mathbf{0}$, that
$\nabla_n \ln b = -g N^{-2} (\alpha_P \Theta_z - \beta_P S_{A_z}) \nabla_n P$, while in the general case we have found in Eq. (65) that $\nabla_n \ln b$ is the sum of these
two contributions. It makes sense that the general case would be the sum of the two special cases since we expect the general
expression for $\nabla_n \ln b$ to be linear in the two relevant epineutral gradients, $\nabla_n \Theta$ and $\nabla_n P$ (consistent with the expansion of $\nabla_p \kappa$
in terms of $\nabla_n \Theta$ and $\nabla_n P$ in going from Eq. (64) to Eq. (65)). But the two special cases have delivered something that the
general case has not, namely that in the two special cases we have been able to find expressions for the integrating factor $b$ in
terms of oceanographic properties on a single vertical water column rather than only in terms of its epineutral gradient through
$\nabla_n \ln b$; recall from Eqs. (55) and (60) that the integrating factor in the two special cases are $\tilde{b}[-gN^{-2}(\ln \tilde{r})_z]$ and
$\tilde{b}[-gN^{-2}(\ln \tilde{\rho}^{\Theta})_z]$ respectively. Given this knowledge we may be tempted to approximate the integrating factor in the general
case as the product of these two expressions that apply in the special cases. That is, an approximation to the integrating factor
might be thought to be

$$b = \frac{\text{NSPV}}{f N^2} = \frac{-g(\ln \gamma)_z}{N^2} \approx \tilde{b} \left[ \frac{-g(\ln \tilde{r})_z}{N^2} \right] \left[ \frac{-g(\ln \tilde{\rho}^{\Theta})_z}{N^2} \right]. \tag{68}$$

This equation is exact at finite amplitude in the two special cases $\nabla_n P = \mathbf{0}$ and $\nabla_n \Theta = \mathbf{0}$, but Dr. Geoff Stanley (personal
communication, 2025) has found that with a reference fluid parcel $(\tilde{S}_A, \tilde{\Theta}, \tilde{P})$ near the equator on a given approximately neutral
surface, this expression is not accurate in either the Southern Ocean or the North Atlantic. Hence, we are not justified in simply
multiplying the two expressions for the integrating factor that apply in the two special case. That is, Eq. (68) is not a valid
approximation to Eq. (66).

The rather simple-looking relationship (64), namely $\nabla_n \ln b = \rho g^2 N^{-2} \nabla_p \kappa$, was published by McDougall (1988), and in
the thirty-seven years since 1988 I have sought a simple physical explanation of it, but without success. The progress reported
here in understanding this relationship is limited to the realization that the terms in $\nabla_n \Theta$ and $\nabla_n P$ that appear on the right-hand
sides (54) and (59) of the two special cases are in fact that the same terms in $\nabla_n \Theta$ and $\nabla_n P$ whose sum is the right-hand side
of the general expression (Eq. 65) for $\nabla_n \ln b$. That is, we can claim to understand the general equation Eq. (66) for $b$ in the



two special cases, and we have been able to find expressions for the integrating factor that avoid the need to perform an
epineutral integral. I would prefer a deeper understanding and faster progress!

## 14 Summary

This article reviews the aspects of physical oceanography in which thermodynamic concepts underlie modern oceanographic practice. These thermodynamic concepts were central to the research that led to TEOS-10 (the international thermodynamic equation of seawater - 2010) and also to the choice of variables that TEOS-10 recommends for oceanographic use. TEOS-10
recognises that Practical Salinity which depends on the electrical conductivity of seawater is affected by the non-standard composition of seawater in a different way than is specific volume. This issue is addressed by TEOS-10 by defining the Absolute Salinity of seawater as the salinity that gives the correct specific volume (at given temperature and pressure). Various methods are given for relating the measured Practical Salinity to the Absolute Salinity that is needed to evaluate density, specific volume and hence the thermal wind relationship.

It is shown that since ocean models to date have made no attempt at including the effects of biogeochemistry, the meridional overturning transports are currently in error by an estimated 13.5% because of this neglect. This percentage figure is poorly known, and this 13.5% figure is for the North Atlantic. Moreover, the salinity variable in both TEOS-10 and EOS-80 based ocean models should be interpreted as Preformed Salinity $S_*$ (or $S_*/u_{PS}$ in the case of EOS-80 models) until at some future time the influence of the non-standard seawater composition is included in these ocean models.

The Fundamental Thermodynamic Relationship and the First Law of Thermodynamics underlie all of seawater thermodynamics, and the derivations of these equations are outlined in this review. We have discussed in detail what is known about the non-conservation of the two main choices that have been used to estimate the heat content of the ocean and have shown that Conservative Temperature closer to being proportional to the heat content of a kilogram of seawater than is potential temperature by two orders of magnitude. We have also discussed the thermodynamic causes of the non-conservation of these
variables.

We have reviewed the thermodynamic reasoning that justifies the neutral plane as being the local surface in which the strong lateral mixing of mesoscale turbulence occurs. Other published alternatives to the neutral tangent plane have also been discussed. The confusing subject of the path-dependent nature of neutral surfaces is also introduced, and some potential implications are discussed. This area of oceanographic research has attracted very little research and very few papers, and any
conclusions drawn above about this topic must be regarded as preliminary; as unfinished business.

*Acknowledgements* This article is based on the 2025 Alfred Wegener Medal lecture "Looking under the hood of Physical Oceanography: Curiosities and Surprises" https://meetingorganizer.copernicus.org/EGU25/sessionprogramme/5775 given at the European Geosciences Union General Assembly in Vienna, 30th April 2025. Valued colleagues Rainer Feistel
(Warnemünde), Richard Pawlowicz (Vancouver) and Paul Barker (Hobart) are thanked for providing unstinting advice and



collaborations over the past almost twenty years. SCOR (the Scientific Committee on Oceanic Research), IAPSO (the International Association for the Physical Sciences of the Oceans) and the University of New South Wales, Sydney, are thanked for providing generous support for this research. I thank Dr. Geoff Stanley for providing valuable comments on section 13. This paper contributes to the tasks of the IAPSO/SCOR/IAPWS Joint Committee on the Properties of Seawater.

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





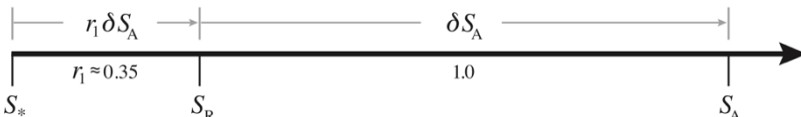

Figure 1: Number line of Salinity, illustrating the differences between Preformed Salinity $S_*$, Reference Salinity $S_R$, and Absolute Salinity $S_A$ for a seawater sample whose composition is different to that of Standard Seawater.

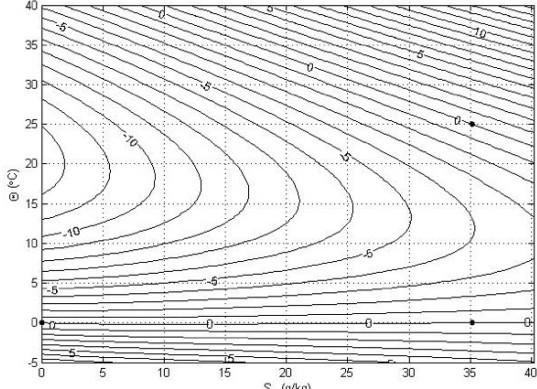

Figure 2: Contours (in °C) of a variable that is used to illustrate the non-conservative production of specific volume at $p = 0$ dbar (where Θ is a conservative variable). The variable is forced to be zero at the three points shown with black dots.

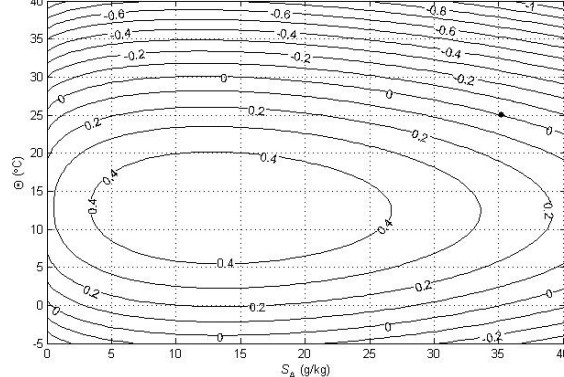

Figure 3: Contours (in °C) of a variable that is used to illustrate the non-conservative production of specific entropy at $p = 0$ dbar (where Θ is a conservative variable).





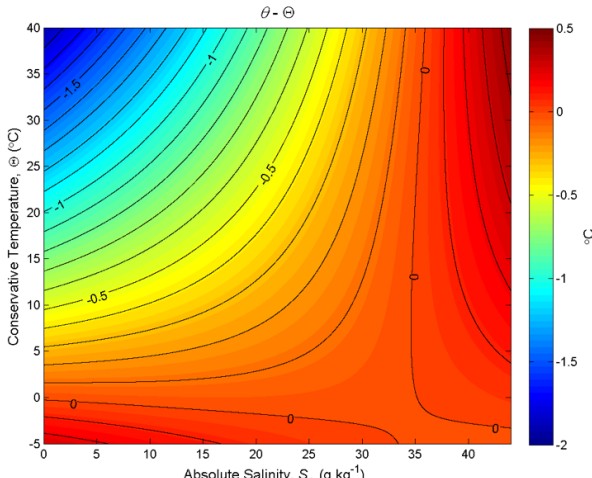

**Figure 4: Contours of the difference between potential temperature and Conservative Temperature, $\theta - \Theta$ (in ℃ ), at $p = 0\mathrm{dbar}$ (where $\Theta$ is a conservative variable). This plot illustrates the non-conservative behaviour of potential temperature.**

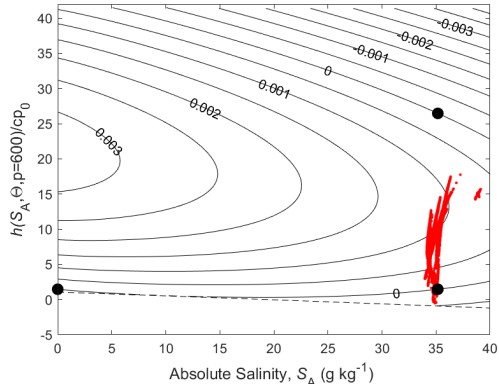

**Figure 5: Contours (in ℃ ) of a variable that is used to illustrate the non-conservative production of Conservative Temperature at $p = 600\mathrm{dbar}$ where $\widehat{h}(S_{A}, \Theta, p = 600\mathrm{dbar})$ is a conservative variable. The variable is forced to be zero at the three points shown with black dots.**



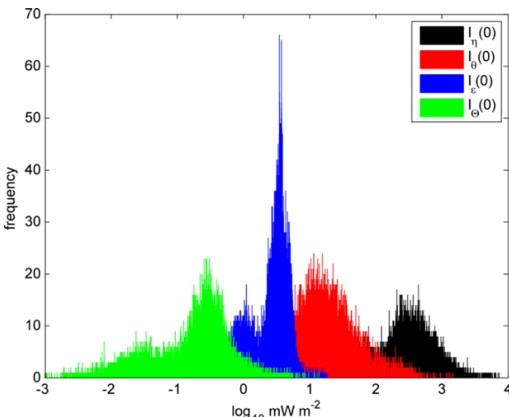

**Figure 6: Histogram of the depth-integrated non-conservative source terms for various variables throughout the world's oceans (from Graham and McDougall, 2013). Five percent of the data points in these histograms exceed $1\ \mathrm{mW\ m^{-2}}$ for Conservative Temperature $\Theta$, $10\ \mathrm{mW\ m^{-2}}$ for the dissipation of turbulent kinetic energy $\varepsilon$, $120\ \mathrm{mW\ m^{-2}}$ for potential temperature $\theta$, and $1200\ \mathrm{mW\ m^{-2}}$ for entropy $\eta$.**

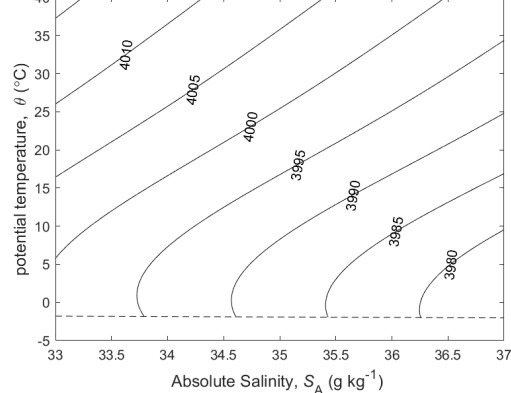

**Figure 7: Contours of the isobaric specific heat capacity $c_p$ (in $\mathrm{J\ kg^{-1}\ K^{-1}}$) of seawater at $p = 0$ dbar.**



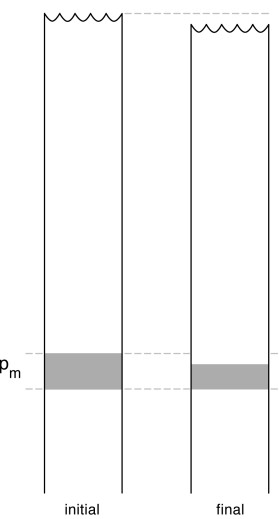

**Figure 8: Diagram illustrating the non-conservation of internal energy and Total Energy (from McDougall et al., 2003). Potential enthalpy referenced to $P^m$ is conserved for parcels on the whole water column, including during the mixing event. At the location of the mixing, both internal energy $u$ and total energy $E$ increase, while specific volume decreases, causing the entire water column above the mixing height to slump downwards. Seawater parcels above the mixing event all have unchanged values of internal energy, enthalpy and potential enthalpy, but they have decreased values of total energy (due to the reduced gravitational potential energy caused by the slumping).**

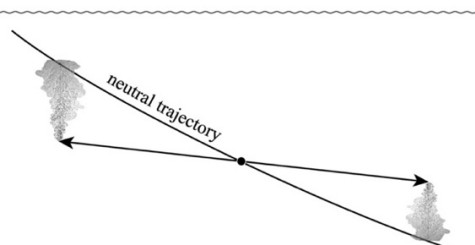

**Figure 9: A sketch of what would be expected if lateral mixing did not occur along the neutral tangent plane (neutral trajectory). If a seawater parcel was moved adiabatically either left or right in a direction that was not neutral, it would find that it had a different specific volume to the ocean fluid at its new location. This difference would drive vertical motion and convection.**





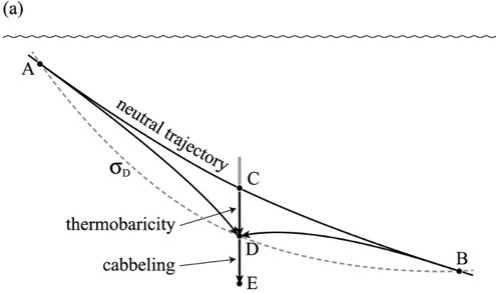

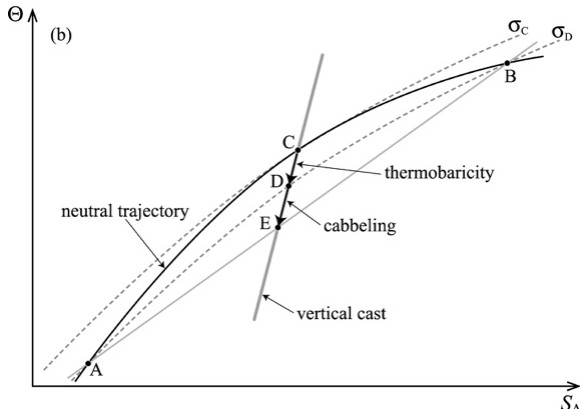

**Figure 10: A sketch used to describe the thermobaricity and cabbeling processes in (a) a vertical cross-section, and (b) on the salinity-temperature diagram.**

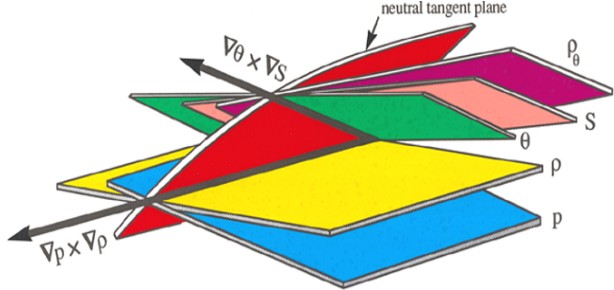

**Figure 11: A sketch showing various planes of constant properties in the vicinity of a given point in the ocean. The neutral tangent plane contains both the lines $\nabla P \times \nabla \rho$ and $\nabla \Theta \times \nabla S_A$. While the neutral tangent plane exists everywhere in space, these little planes do not link up to form a well-defined neutral surface unless the neutral helicity is zero everywhere. This requires that the two lines $\nabla P \times \nabla \rho$ and $\nabla \Theta \times \nabla S_A$ coincide.**



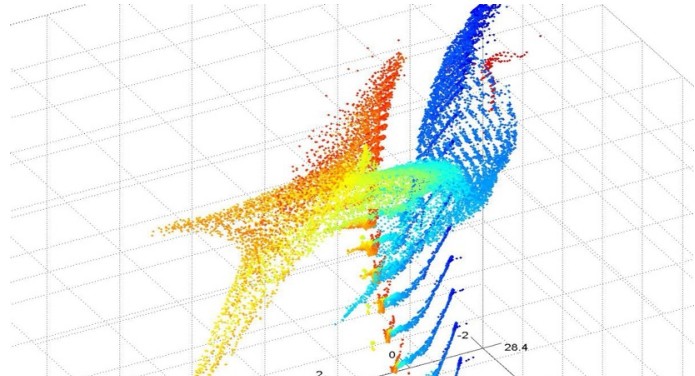

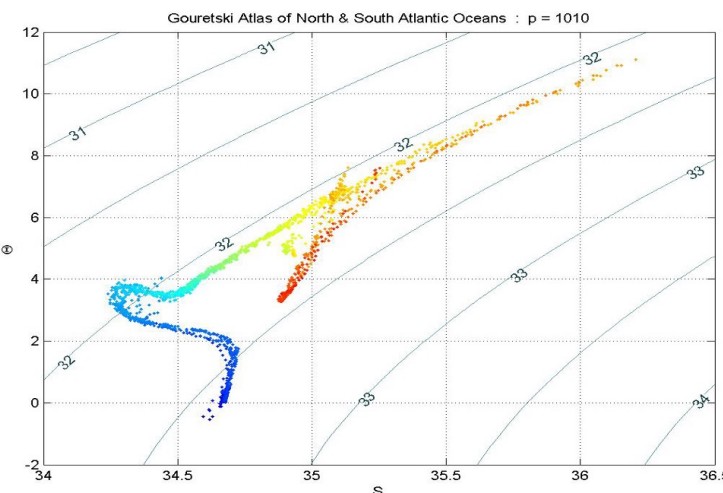

**Figure 12: (a) A view of the South and North Atlantic hydrographic data in three-dimensional $(S_A, \Theta, P)$ space. (b) A section through**
**the same $(S_A, \Theta, P)$ data at a pressure of 1,010 dbar.**



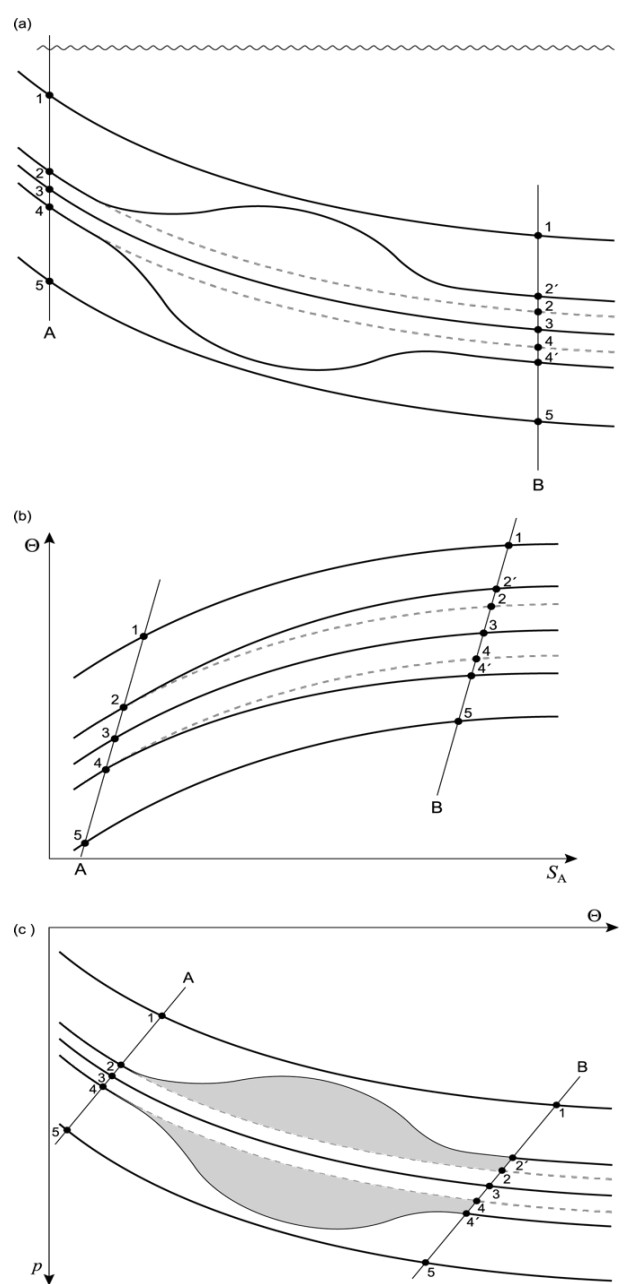

**Figure 13: An ocean cross-section illustrating aspects of the epineutral variations of the integrating factor *b*.**