# Peer review of "Thermodynamic Concepts used in Physical Oceanography"

_EGUsphere, 2025_

## Referee Comment (RC1)

**Review of McDougall: Thermodynamic concepts in physical oceanography**

Review provided by Stephen Griffies Princeton University October 22, 2025

**1 Summary**

This is a useful review of thermodynamics used in ocean physics, drawn from the author's published work and based on his 2025 EGU Wegener Medal lecture. The author is the world's leader in ocean thermodynamics and its applications. I offer only some minor comments focused on points of clarification. So an official recommendation is "minor revision."

**2 Specific comments**

**Why salinity at the start of the manuscript?**

It is not clear to me why the salinity section is the lead section after the Introduction. There are few concepts here. Instead, it details various recipes and papers where rationales are presented. I suggest moving this section to later in the paper, to thus start with the major conceptual piece concerned with the First Law.

**line 116**

Explain "SI-traceable manner".

**line 116**

I suggest adding a few words to indicate where  $u_{PS} = 35.16504/35$  comes from in equation (1). At the least, provide a few conceptual bread crumbs for the interested reader.

**line 165**

"total derivatives"  $\rightarrow$  "total differentials" (as correctly used on line 169).

**intensive variables in equation (4)**

Throughout this manuscript, h, u, and  $\eta$  are referred to as enthalpy, internal energy, and entropy. In fact, these terms are all per unit mass. This point should be noted around equation (4), the fundamental thermodynamic relation, FTR,

$$dh - v dP = du + P dv = T d\eta + \mu dS_A.$$
(1)

**lines 169-170: reversible versus quasi-static**

The fundamental thermodynamic relation holds between two thermodynamic equilibria. On line 169-170, McDougall further states that the total differentials represent reversible differences between such equilibria. A similar statement was also present in an early version of McDougall et al. (2023). However, as I noted in my peer-review of that paper, the physics literature, such as the gold standard texts from Reif (1965) and Callen (1985), allow for the FTR to hold for the broader category of processes known as quasi-static. Consequently, the FTR, in their presentation, holds for both reversible and irreversible quasi-static processes. I note that earlier treatments of thermodynamics did not make the distinction, but the distinction became important when more work was done to understand irreversible processes in the 20th century (Callen's thesis

work in 1947 was on irreversibility). As a result, we can extend the FTR to include friction and diffusion, which is clearly important for ocean physicists to use the FTR in a moving and real fluid.

In response to this earlier reviewer comment, *McDougall et al.* (2023) added a nice discussion in their Section 1.2 about some basic conceptual issues. However, no where in that discussion did they argue for the FTR holding just for reversible processes rather than the broader class of quasi-static processes. They focus on the need for thermodynamic state functions, in particular entropy, to remain state functions when working with a moving fluid, thus allowing for differentials to transfer over to material time derivatives. But these properties are precisely the properties maintained by quasi-static processes.

So in summary, the present manuscript, with the word "conceptual" front and center in the title, is an ideal place to raise these conceptual points. How we in the ocean physics community make use of the First Law has a lot to do with the hypothesis of local thermodynamic equilibrium (Section III.2 of *DeGroot and Mazur* (1984)) and the distinction between quasi-static versus reversible processes. That is, the present paper is just the place to address, head-on, what *McDougall et al.* (2023) refer to as "otherwise annoying conceptual issues", which were sidestepped in that paper.

**line 171**

"We tend to regard" is a rather qualified and tentative language. Is there a reason for such language?

**line 192**

I am puzzled why the kinetic energy per mass is not afforded its own symbol, such as

$$K = \frac{1}{2}\mathbf{u} \cdot \mathbf{u}.\tag{2}$$

The current use of  $0.5 \mathbf{u} \cdot \mathbf{u}$  throughout the energetic discussions is clumsy.

**line 197**

I suggest dropping "quality" from the following sentence, since it implies that those books not treating the First Law have low quality.

"This evolution equation is derived in detail in quality fluid dynamics textbooks..."

Namely there are any number of fluid dynamics texts that are only concerned with hydrodynamics, waves, vorticity, etc., and choose not to treat thermodynamics. I do not think you wish to imply that all such books are low quality.

**lines 236-239**

These lines are rather packed with ideas yet not supported by the maths. I think it useful to include some equations. Additionally, wording in these lines can lead to confusion. Namely, molecular diffusion vanishes in thermodynamic equilibrium since *in situ* temperature and chemical potential are uniform. But the way it is written, it seems one needs to assume zero molecular diffusion in order to realize thermodynamic equilibrium. Some rewording would be useful.

I recommend adding some of the text from the TEOS-10 manual to clearly explain why thermodynamic equilibrium is irrelevant for the ocean, given flow has a turbulent nature and so homogenizes potential properties rather than in situ temperature and chemical potential. This point is only hinted at here, by noting that a thermodynamically equilibrated ocean has a huge and unobserved vertical gradient of  $S_A$ . But there is no statement to the effect that "the observed ocean is no where near thermodynamic equilibrium for reasons A and B." Instead, the reader is meant to infer that conclusion, which is not a good strategy for a review paper aimed at a broad audience.

**lines 269 and 277**

The right hand side of equation (12) has  $-\nabla \cdot \mathbf{F}^C$ . With the minus sign this term represents the "convergence of the flux,  $\mathbf{F}^C$ ." Hence, I suggest calling it the flux convergence rather than the flux divergence.

**Arguments surrounding equation (16)**

The discussion leading up to equation (16) makes use of some standard Leibniz-Reynolds arguments for finite volume budgets. However, I am puzzled by details of the chosen control volume used to throw away the boundary integral term while maintaining constant pressure,  $P = P^m$ . A picture of the control volume would prove very useful here.

In particular, the restriction  $P = P^m$  needed for equation (15) means we are restricting attention to the isobaric surface. So when performing a three-dimensional finite volume budget, how can all of the region remain with  $P = P^m$ , unless you are assuming constant pressure within the volume, which seems odd. For these reasons, I am unsure why we can drop the  $\mathbf{u}^{\text{dia}}$  term projected onto the normal direction of the boundaries. I think a carefully drawn schematic of the finite volume would help heaps.

Also, I do not understand why molecular mixing is absent (line 301). Is it instead neglected due to dominance by turbulence?

So the bottom line question I have is the conclusion on line 304. Namely, the arguments leading to equation (16) are finite volume arguments in 3-dimensions, whereas turbulent mixing at  $P = P^m$  is a local process that occurs at a point or on a surface.

**line 305**

"The almost conservative behaviour of  $h^m$ ...". Where in the previous text is it shown that  $h^m$  is almost conservative rather than fully conservative?

**Equation (18) and following**

The one-sentence paragraph ending with equation (18) introduces  $\hat{h}_{\Theta}$  and  $\hat{\eta}_{\Theta}$ , yet without defining the hats and the subscripts. I presume the subscripts are partial derivatives, but that needs to be stated. But what are the hats for?

My guess is that this section is a cut-and-paste from another document, with incomplete transfer of symbols. Indeed, on line 336 we find a reference to equation (R.02), but that equation is no where in the present manuscript.

**Equation (23)**

Again we find the undefined symbols  $\hat{h}$  and  $\hat{\eta}$ .

**line 461**

Need to define  $T_0 + t$ . I note that it is unfortunate that t is used here for a temperature whereas it is used earlier for time.

**line 474**

It would be useful to give numbers for the heating due to dissipation of turbulent kinetic energy, in both the ocean interior and in boundary layers. Also, state why we in the ocean physics community can ignore this dissipation when modeling the large-scale circulation, whereas atmospheric modelers find it essential.

**line 483**

It is useful to remind the reader that mixing occurs at constant pressure, which then indicates why enthalpy is conserved during mixing.

**line 491**

The second  $m_1$  should be  $m_2$ .

**line 519**

spelling "derivates"

**line 523**

evaluate  $\rightarrow$  evaluated

**line 685**

"in terms of"

**line 742**

I suggest " $2\frac{1}{2}$ %" is better written as "2.5%"

**line 815**

Need to add "= 0" after " $\beta \delta S_A - \alpha \delta \Theta$ ."

**line 903**

"...where the over-hats indicate that these variables are functions of  $(S_A, \Theta, P)$ ." Does this specification also hold for all other hat variables in the manuscript that were written earlier? If so, then the reader would welcome having that information stated way back near equation (18).

**line 944**

The sentence starting "In that case..." is awkward. I suggest rewording.

**Section 13**

Section 13 is title "Neutral Surface Potential Vorticity," which is the title of *McDougall* (1988). However, as noted early in Section 13, we are here only concerned with planetary geostrophic potential vorticity. I thus recommend changing the section title to "Neutral Surface Planetary Geostrophic Potential Vorticity." Although a bit more of a mouth-full, it does serve as a more honest advertisement for what is to come.

**Expression for potential vorticity**

Is the neutral surface potential vorticity defined by equation (48) materially invariant in the absence of irreversible processes? Was this shown in McDougall (1988)?

**line 1113**

Reference is made to equation (113). This manuscript has no such equation. Perhaps another cut-and-paste error.

**Equation (61)**

This reviewer admits being a bit exhausted by this point, so I can certainly accept if I missed something. But what exactly is  $\kappa$ ? That math symbol doesn not appear in equation (47), which is claimed to have its curl render equation (61). Perhaps it is defined earlier and I forgot where?

**Figures**

It is a shame the figures were not inserted in their proper place, thus making the reading far simpler. The old days of relegating figures to the end of the preprint seem unnecessary in today's world, particularly when few readers actually print a hard copy.

Some of the figures (e.g., 2, 3, 11) do not render well when zooming. Perhaps they are taken from their original publications from decades ago. It would be a shame to have blurry figures in this otherwise thorough and impressive review.

**Figure 2**

I have a tough time finding the black dots.

**References**

- Callen, H. B., Thermodynamics and an Introduction to Thermostatics, John Wiley and Sons, New York, 493 + xvi pp, 1985.
- DeGroot, S. R., and P. Mazur, *Non-Equilibrium Thermodynamics*, Dover Publications, New York, 510 pp, 1984.
- McDougall, T. J., Neutral-surface potential vorticity, *Progress in Oceanography*, 20, 185–221, doi:10.1016/0079-6611(88)90002-X, 1988.
- McDougall, T. J., P. M. Barker, R. Feistel, and F. Roquet, A thermodynamic potential of seawater in terms of Absolute Salinity, Conservative Temperature, and *in situ* pressure, *Ocean Science*, 19, 1719–1741, doi:10.5194/os-19-1719-2023, 2023.
- Reif, F., Fundamentals of Statistical and Thermal Physics, McGraw-Hill, New York, 1965.